# PPARγ lipodystrophy mutants reveal inter-molecular interactions required for enhancer activation

Maria Stahl Madsen [1,14], Marjoleine F. Broekema [2,10,14], Martin Rønn Madsen[1,11], Arjen Koppen [2], Anouska Borgman[2], Cathrin Gräwe [3], Elisabeth G. K. Thomsen [4], Denise Westland[2], Mariette E. G. Kranendonk[2,12], Marian Groot Koerkamp[2,12], Nicole Hamers[2], Alexandre M. J. J. Bonvin [5], José M. Ramos Pittol [2,13], Kedar Nath Natarajan [1], Sander Kersten [6], Frank C. P. Holstege [2,12], Houshang Monajemi [7,8], Saskia W. C. van Mil [2], Michiel Vermeulen [3], Birthe B. Kragelund [4], David Cassiman [9], Susanne Mandrup [1,15] ✉ & Eric Kalkhoven [2,15] ✉

Peroxisome proliferator-activated receptor γ (PPARγ) is the master regulator of adipocyte differentiation, and mutations that interfere with its function cause lipodystrophy. PPARγ is a highly modular protein, and structural studies indicate that PPARγ domains engage in several intra- and inter-molecular interactions. How these interactions modulate PPARγ's ability to activate target genes in a cellular context is currently poorly understood. Here we take advantage of two previously uncharacterized lipodystrophy mutations, R212Q and E379K, that are predicted to interfere with the interaction of the hinge of PPARγ with DNA and with the interaction of PPARγ ligand binding domain (LBD) with the DNA-binding domain (DBD) of the retinoid X receptor, respectively. Using biochemical and genome-wide approaches we show that these mutations impair PPARγ function on an overlapping subset of target enhancers. The hinge region-DNA interaction appears mostly important for binding and remodelling of target enhancers in inaccessible chromatin, whereas the PPARγ-LBD:RXR-DBD interface stabilizes the PPARγ:RXR:DNA ternary complex. Our data demonstrate how in-depth analyses of lipodystrophy mutants can unravel molecular mechanisms of PPARγ function.

The nuclear receptor peroxisome proliferator-activated receptor γ (PPARγ), encoded by the *PPARG* gene, is the master regulator of adipocyte differentiation and function and an important regulator of whole-body lipid metabolism and insulin sensitivity[1]. Agonists include several unsaturated fatty acids and lipid metabolites as well as various synthetic compounds such as insulin-sensitizing thiazolidinediones. The importance of PPARγ in human adipocyte biology and physiology is underscored by the finding that many cases of familial lipodystrophy, a syndrome characterized by repartitioning of adipose tissue

causing severe insulin resistance, type 2 diabetes mellitus (T2DM) and dyslipidemia, are caused by heterozygous point mutations in this transcription factor[2] (FPLD3; OMIM 604367).

PPARγ regulates transcription of its target genes by binding as a heterodimer with members of the retinoid X receptor (RXR) subfamily to PPAR-response elements (PPREs), which are degenerate repeats of 5′-AGGTCA-3′ spaced by one nucleotide[3,4]. Here RXR occupies the 3′ half site, while PPARγ binds to the 5′ half site and its 5′ extension[5]. In adipocytes, the PPARγ:RXR heterodimer cooperates with the CCAAT/

enhancer-binding protein α (C/EBPα) in the activation of many target genes[1]. PPARγ and C/EBPα bind to many of the same target enhancers, in some cases in a highly interdependent manner. For most of these interdependent binding sites, C/EBPα acts as the leading transcription factor facilitating PPARγ binding; however, PPARγ can also act as a leading factor facilitating C/EBPα binding[6].

Similar to other nuclear receptors, PPARγ is a modular protein composed of two highly structured and evolutionary conserved domains, the DNA-binding domain (DBD) and the ligand-binding domain (LBD), as well as two mostly unstructured and less conserved domains, the N-terminal A/B-domain and the hinge region[7]. While in vitro experiments indicate that the different domains of PPARγ can execute their functions independently, several results point to intra- and intermolecular interactions being important for transactivation by the full-length protein[7–9]. Early cell-based studies indicate intramolecular communication between the ligand-dependent transactivation

function in the LBD and the ligand-independent transactivation function in the N-terminus and showed that phosphorylation of S112 in the N-terminal domain can affect ligand binding[10]. Furthermore, studies of the structure of the DNA-bound PPARγ:RXR heterodimer using either X-ray crystallography or small-angle X-ray scattering (SAXS) indicate that several different intermolecular interfaces (DNA-protein and protein-protein) are directly involved in regulating PPARγ activity[11–14]. Thus, the structures indicate that the hinge domain of PPARγ contributes to specificity in DNA-binding by making contacts to the bases in the minor groove of the 5'extension of the PPRE[11–13]. Furthermore, these studies also indicate that heterodimerization involves several domains of both PPARγ and RXR. X-ray crystallography[11] and SAXS[14] revealed a compact, "closed" conformation of the DNA-bound heterodimer with three dimerization interfaces involving (i) the RXRα LBD (helix 10) and the PPARγ LBD (helix 10 and 11); (ii) the RXRα DBD and the PPARγ DBD (bridged by the DNA); and (iii) the RXRα DBD and the

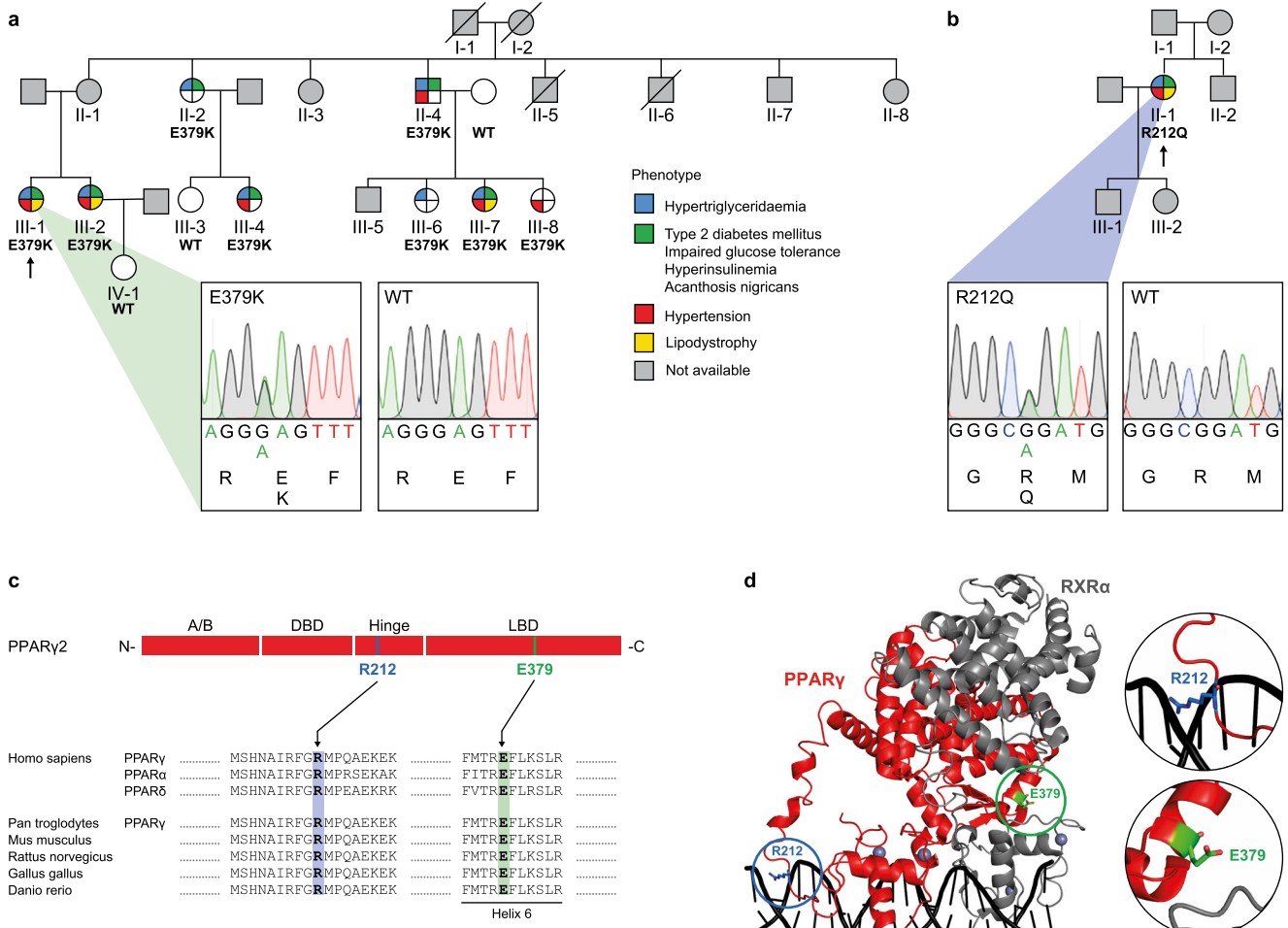

**Fig. 1 | Identification of PPARγ2 E379K and R212Q. a** Family pedigree of index patient 1. Each family member is numbered for identification. The proband is indicated by an arrow. Squares and circles indicate males and females, respectively. Phenotypes are elaborated by color segments showing the presence of specific features. Gray symbols denote individuals that were not available for DNA analysis. Deceased individuals are indicated by a diagonal line through the symbol. DNA sequence analysis showing the heterozygous E379K mutation. The chromatogram shows both alleles from the patient (left panel) in comparison with corresponding genomic DNA from a non-affected individual (right panel). For tracing, the nucleotide and amino acid sequences are shown. **b** Family pedigree of index patient 2, harboring a heterozygous R212Q mutation. See description of panel a for details on representation. **c** Top: Schematic representation of domains in PPARγ2; N-terminal A/B-domain, DNA-binding domain (DBD), hinge region, and ligand-binding domain (LBD) and indicated positions of the two mutations. Bottom: Alignment of the amino acid sequence surrounding PPARγ2 E379K and R212Q between human PPAR subtypes and PPARγ between different species. Residue positions of E379 and R212 are highlighted in green and blue, respectively. **d** Crystal structure of PPARγ:RXRα heterodimer bound to DNA (PPARγ in red; RXRα in gray; PDB entry 3DZY)[11]. E379 (in green) at the end of helix 6 in PPARγ at the heterodimerization interface with RXRα DBD and R212 (in blue) in the hinge region of PPARγ are encircled. Both amino acid residues are indicated in stick format. Protein Database entry 3DZY. The figure is generated by open-source software PyMOL2 (www.pymol.org). A similar DNA-bound conformation based on SAXS was proposed by Bernardes et al.[14].

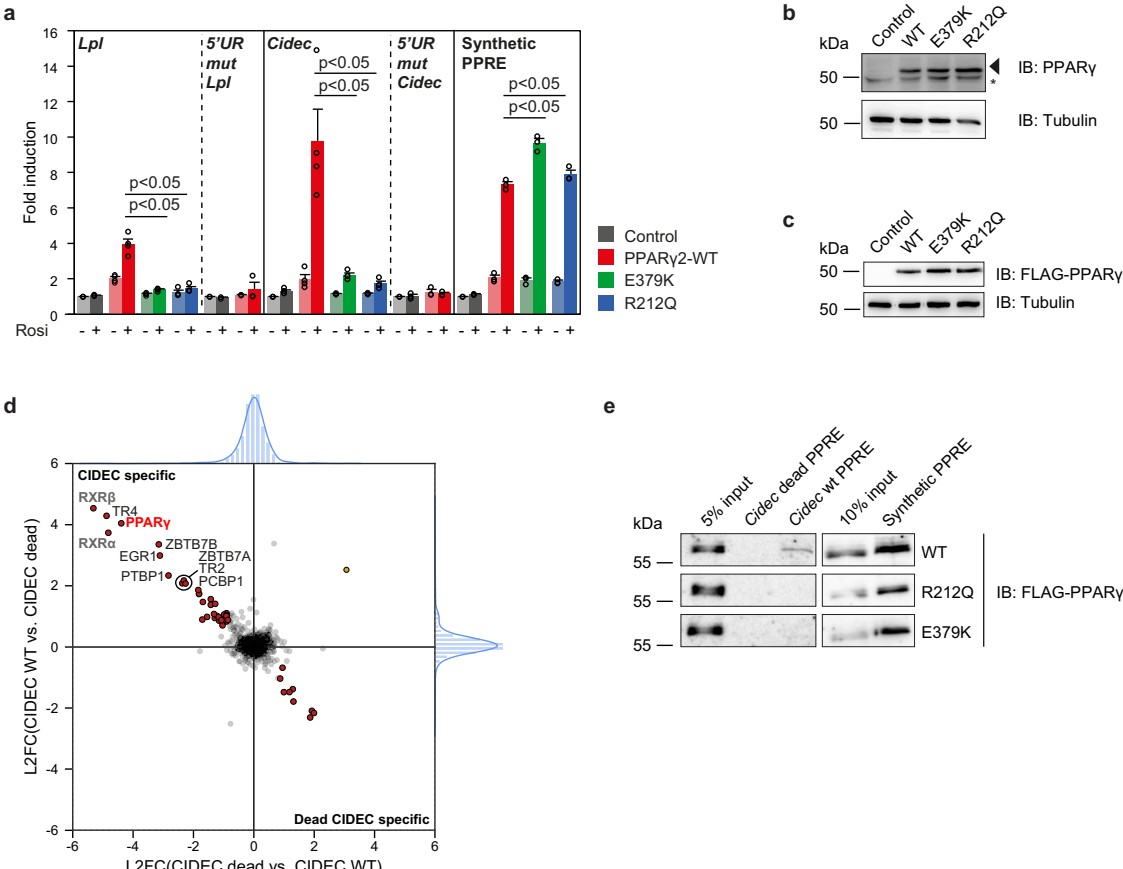

**Fig. 2 | E379K and R212Q mutants destabilize PPARγ:RXR binding to DNA in vitro. a** U2OS cells were transiently cotransfected with expression vectors encoding PPARγ-WT or mutants and different reporter constructs as indicated, in the absence or presence of 1 μM rosiglitazone. Activation of the reporter is expressed as fold induction over that with empty vector (control). Data are presented as mean values + SEM, with individual data points indicated with circles, $n = 3–4$ biologically independent experiments. One-way ANOVA with Tukey's multiple comparisons were used to compare cells transfected with mutant vs. WT; *$p < 0.05$ cells transfected with mutant vs. WT. **b** Expression of the different PPARγ proteins transiently overexpressed in U2OS cells, as assessed by western blot. The arrow indicates PPARγ, and the asterisk indicates a non-specific band. Control, empty vector control; WT, PPARγ wild-type. Three independent experiments were performed and similar results were obtained. **c** Expression of the different FLAG-tagged PPARγ proteins stably overexpressed in U2OS cells, as assessed by western blot using a FLAG-tag antibody. Control, empty vector control; WT, wild-type.

Three independent experiments were performed and similar results were obtained. **d** DNA affinity purification-mass spectrometry analysis of *Cidec* PPRE interactors. Forward and reverse experiments were performed using oligonucleotides containing the *Cidec* PPRE motif or a mutant version (*Cidec* dead), followed by dimethyl labeling and mass spectrometry analysis. Log2 ratios (L2FC) of all identified and quantified proteins (from nuclear extracts) in both experiments were plotted against each other. Proteins binding equally well to both oligonucleotides center around log2(ratio) = 0 and are marked in light gray. Proteins binding significantly better to the *Cidec* PPRE motif or the *Cidec* dead motif were determined by outlier statistics. These proteins are marked in red. **e** DNA affinity purification followed by western blot analysis were performed using oligonucleotides containing the *Cidec* PPRE motif, the *Cidec* dead motif and the synthetic PPRE motif. Pulldowns were performed using nuclear extracts containing the different FLAG-tagged PPARγ proteins. Three independent experiments were performed, and similar results were obtained. Source data for panel **a**–**c** and **e** are provided in the Source Data file.

PPARγ LBD (helix 6). This latter PPARγ:RXR LBD-DBD interaction, which was supported by hydrogen-deuterium (H/D) exchange mass spectrometry[11,14] and mutagenesis studies[11], may critically depend on DNA binding, as it was not observed in the absence of DNA[14]. Furthermore, it should be noted that a more elongated, "open" conformation of the DNA-bound heterodimer lacking the PPARγ:RXR LBD-DBD interaction was reported in other SAXS studies, in this case supported by FRET studies[12,13]. More recently, LBD-DBD interdomain interactions were supported by analogy with the RARβ:RXR heterodimer[9]. Regardless of these differences in structural approaches and data interpretation, it is currently poorly understood how complex intermolecular interactions may modulate the ability of PPARγ to activate target enhancers in cooperation with other transcription factors in the context of chromatin.

Here, we investigate the genome-wide epigenomic and transcriptional effects of two previously uncharacterized FPLD3-associated *PPARG* mutations, i.e., PPARγ-R212Q in the hinge region and PPARγ-E379K in the LBD, both predicted to interfere with

intermolecular interactions of the ternary PPARγ:RXR:DNA complex. We show that both mutations impair the adipogenic capacity of PPARγ as well as the activation of an overlapping subset of enhancers that are characterized by being highly dependent on PPARγ for chromatin remodeling. These findings provide mechanistic insights into the function of PPARγ and emphasize the importance of PPARγ as the leading transcription factor in a subset of target enhancers.

## Results

### Identification of the FPLD3-associated PPARγ mutations E379K and R212Q

FPLD3-associated *PPARG* mutations present a valuable tool to unravel the complex intermolecular interactions (both protein-protein and protein-DNA) required for optimal enhancer activation by PPARγ as they invariably result in loss of function[2,15]. We selected two previously uncharacterized FPLD3-associated *PPARG* mutations that are both predicted to affect intermolecular interactions based on structural studies. The first represents a novel heterozygous

missense FPLD3 mutation (Fig. 1a), substituting a highly conserved glutamic acid at position 379 in helix 6 in the LBD of PPARγ with a lysine (E379K; Fig. 1a, c). Genotyping showed the same mutation in seven additional family members, all of whom have derangements in lipid and glucose metabolism, whereas it was absent in family members without metabolic derangements (Fig. 1a). Crystallography and SAXS of the DNA-bound PPARγ:RXRα heterodimer have indicated that E379 is located at the heterodimerization interface between PPARγ LBD and RXRα DBD (Fig. 1d)[11,14]. An alternative conformation has been proposed in other SAXS studies, where E379 points outwards from the ternary complex (Supplementary Fig. 1)[12].

The second heterozygous mutation we selected results in the substitution of a highly conserved arginine with glutamine (R212Q; Fig. 1b, c). Also, in this case, we identified this mutation in a patient with clear lipodystrophy features (Fig. 1b). Germline transmission could not be established as family members were unavailable for genotyping, but R212Q and R212W have previously been reported in FPLD3 patients by others[15,16]. Arginine 212 is located in the hinge region and forms interactions with the minor groove of the DNA helix immediately 5′ of the PPRE (referred to as the 5′ extension or 5′ upstream region; 5′ UR), and substitution of arginine with a glutamine residue may reduce these (Fig. 1d and Supplementary Fig. 1)[11,12].

Taken together, we identified two previously uncharacterized lipodystrophy mutations that are predicted to interfere with two interaction interfaces of PPARγ that remain functionally poorly understood.

### E379K and R212Q mutants destabilize PPARγ:RXR binding to DNA in vitro

The fact that E379 and R212 residues are located in predicted, but functionally elusive interaction interfaces of PPARγ:RXR and PPARγ:DNA, respectively, prompted us to study the transcriptional and epigenomic effects of the E379K and R212Q lipodystrophy mutants in detail. We first investigated the effect of these two mutations on the ability of PPARγ to activate transcription in reporter assays in U2OS cells, which express negligible levels of endogenous PPARγ[17]. Cells were transiently transfected with expression plasmids encoding WT or mutant PPARγ and reporter constructs regulated by the well-established PPRE located in the promoter-proximal enhancers of the lipoprotein lipase (Lpl) gene[3,18] and the Cell Death Inducing DFFA Like Effector C gene (Cidec; also referred to as Fsp27)[3,19] (Supplementary Fig. 2). Both PPREs are imperfect repeats with partly conserved 5′extensions (Supplementary Fig. 2). Compared to WT PPARγ, the two mutants display greatly reduced ability to activate both reporter constructs (Fig. 2a) at comparable levels of expression (Fig. 2b). In agreement with previous data[15], the FPLD3-associated R212W mutant also displays reduced activity (data not shown), similar to the R212Q mutant reported here. Consistent with the importance of the 5′ extension for imperfect PPREs[5,11,12], WT PPARγ loses its ability to activate the Lpl and Cidec PPRE reporters when this sequence is mutated (Fig. 2a). Interestingly, however, both mutants readily activate a reporter containing a synthetic, perfectly palindromic PPRE identical to the one used for X-ray and SAXS structural studies[5,11,12]. These results indicate that the R212Q mutation, as well as E379K mutation, destabilizes the DNA-binding of PPARγ at imperfect PPREs.

To specifically compare the DNA-binding properties of the WT and mutant PPARγ proteins, we performed DNA pulldown assays. Nuclear extracts from U2OS cells stably expressing comparable levels of the different PPARγ proteins (Fig. 2c) were incubated and pulled down with the Cidec PPRE, a mutated Cidec PPRE, or the synthetic PPRE. Mass spectrometry analysis confirmed that the WT Cidec PPRE specifically pulled down PPARγ and RXRα and -β (Fig. 2d), and western blotting showed that DNA binding of mutants is severely compromised on the Cidec PPRE but not on the synthetic PPRE (Fig. 2e). In support of the E379 residue being important in the DNA-bound state only[14],

heterodimerization in the absence of DNA, which is known to critically depend on the PPARγ:RXR LBD-LBD interface[17], was intact (Supplementary Fig. 3a–e). Furthermore, when tested in isolation, the PPARγ:RXR LBD-DBD interface was insufficient for efficient heterodimerization in vitro (Supplementary Fig. 3d). Taken together, these results indicate that although neither mutation affects the DBD, they both interfere with in vitro DNA binding on natural imperfect PPREs.

The finding that the R212Q mutant interferes with DNA binding is consistent with the predicted role of the N-terminal part of the hinge region in DNA binding to the 5′ extension of the PPRE[11,12] (Fig. 1d and Supplementary Fig. 1). However, the E379 residue is located in the LBD and does not directly contact the DNA (Fig. 1d and Supplementary Fig. 1). Close inspection of the PPARγ:RXRα crystal structure shows a complex inter- and intramolecular interaction network involving a quartet of residues, where E379 interacts with a tyrosine in RXRα (Y189), with stabilizing intra-helical interactions between E379 and K382 in helix 6 of PPARγ and an intramolecular interaction between Y189 and K175 in RXRα (Fig. 3a). The importance of the stabilizing interaction within helix 6 was supported with spectroscopic NMR and CD data of WT and E379K peptides, with severe loss of helicity in the E379K peptide (Fig. 3b). Therefore, the E379K mutant is predicted to perturb this specific interaction interface with RXRα and is expected to lead to destabilization of the heterodimer:DNA complex. In silico predictions based on the crystal structure indicated a potential for compensation by an artificial charge reversal RXRα-K175E mutant, which would restore the important contact between RXR and helix 6 through a new electrostatic interaction between E379K$_{PPARγ}$ and K175E$_{RXRα}$ (Fig. 3a). Experimental testing of this model in HEK293T-cells, which express negligible levels of endogenous PPARγ and RXRα[15] revealed that the RXRα-K175E mutant displays a similar transcriptional defect as the PPARγ-E379K mutant (Fig. 3c, d). Interestingly, however, the combination of the two mutants rescued activation of the Lpl and the Cidec reporter (Fig. 3c, d and Supplementary Fig. 4), suggesting that the new PPARγ-K379/RXRα-E175 salt bridge can restore a functional PPARγ:RXR heterodimer. We wished to exclude additional functional defects of the E379K mutation and focused on ligand-dependent coregulator docking and subsequent transcriptional activation, two intertwined functions that are also mediated by the LBD and affected in several natural PPARγ mutants (e.g., refs. 20, 21). When testing the LBD in isolation, no clear functional defects of the E379K mutation were observed regarding ligand-dependent in vitro coregulator binding and transcriptional activity (Supplementary Fig. 5).

In summary, the PPARγ lipodystrophy-associated mutations R212Q and E379K, located in the hinge and LBD, respectively, both markedly destabilize the binding of the PPARγ:RXR heterodimer to natural imperfect PPREs but through effects on different interfaces.

### E379K and R212Q mutants impair adipogenic capacity of PPARγ

To investigate how these mutations affect the epigenomic and transcriptional effects of PPARγ in chromatin, we took advantage of our recently developed model system based on immortalized PPARγ$^{-/-}$ mouse embryonic fibroblasts stably transduced to express high levels of coxakie adenoviral receptor (CAR) for efficient uptake of adenoviruses (PPARγ$^{-/-}$ MEF-CARs)[6]. We have previously shown that short-term exposure of these cells to adenoviruses expressing PPARγ leads to the activation of PPARγ-target enhancers and induction of many adipocyte genes. We, therefore, introduced the R212Q and E379K mutations into the corresponding sites in murine PPARγ and titrated the adenoviral vectors to express similar levels of WT and mutant PPARγ in PPARγ$^{-/-}$ MEF-CAR cells (Fig. 4a, b). Following 2 h of exposure to adenovirus, the medium was removed, and cells were stimulated with an adipogenic cocktail. Interestingly, after seven days of incubation, cells transduced with WT PPARγ accumulate lipids and express adipocyte marker genes, whereas differentiation is much less efficient in cells transduced with the mutant PPARs (Fig. 4c, d). These findings

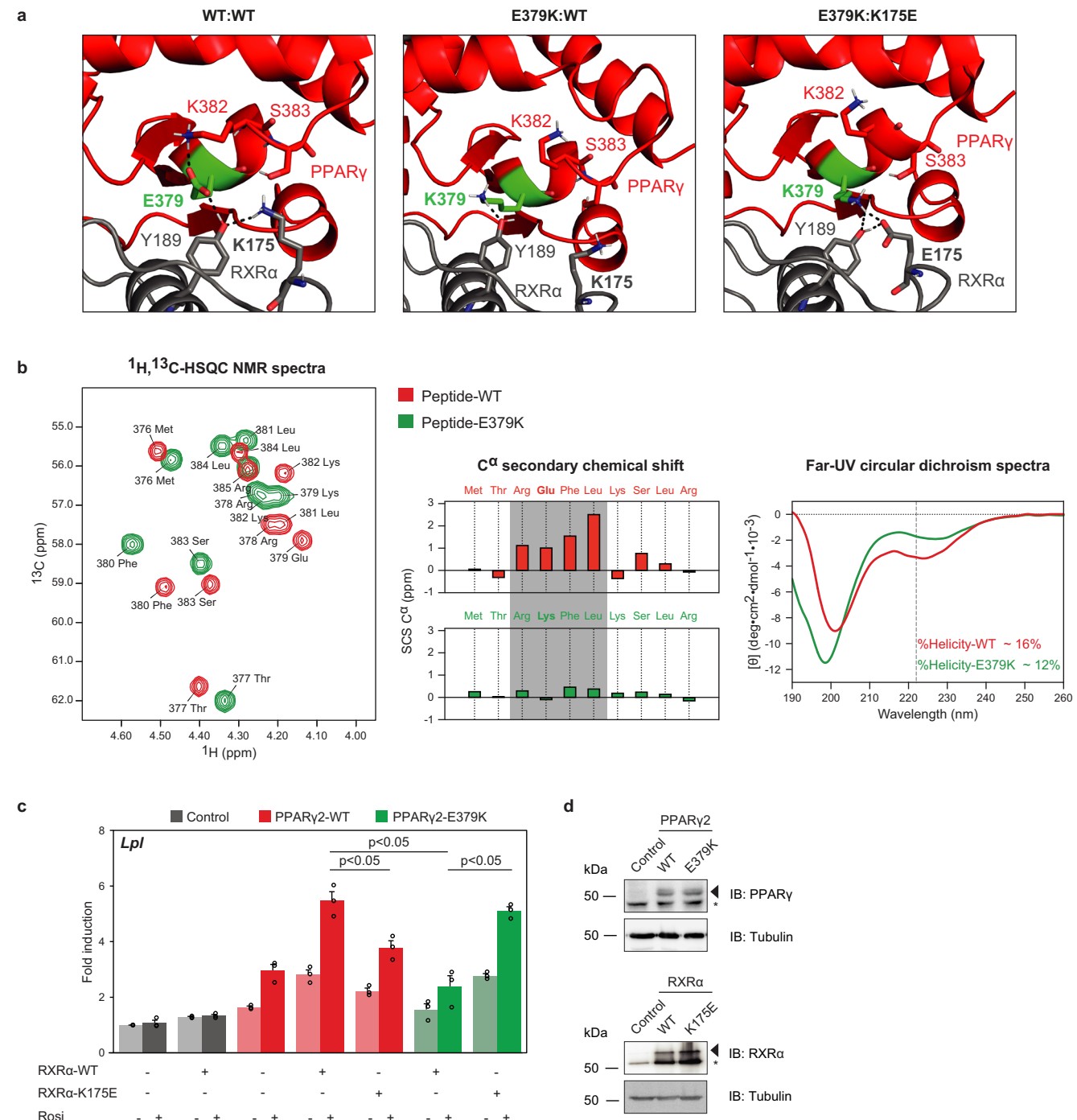

indicate that the ability of PPARγ to drive adipocyte differentiation is impaired by both mutations but most markedly by the R212Q mutation.

### E379K and R212Q mutants compromise acute activation of a subset of target genes

To obtain molecular insights into the mechanisms underlying the reduced adipogenic potential of the PPARγ mutants (Fig. 4), we compared the ability of adenovirally expressed WT and mutant PPARγ to acutely activate gene expression in PPARγ[-/-] MEF-CAR cells in the presence of rosiglitazone (Fig. 5a). In this setting, PPARγ-WT expression changes the expression of 399 genes (DESeq2, false discovery rate (FDR) < 5% and fold change PPARγ-WT vs Control ≥ 1.5 or ≤ −1.5), with 277 genes being induced and 122 genes being repressed compared to

control cells (Fig. 5b and Supplementary Tables 2, 3). For further analyses, we focused on the genes induced by PPARγ-WT, as the mechanisms underlying PPARγ-mediated gene repression are less well understood and likely indirect[22,23]. As expected, genes induced by PPARγ-WT are enriched in adipocyte-related GO-categories (e.g., lipid droplet organization, lipid storage, cellular triglyceride homeostasis, fat cell differentiation), and genes that contribute most to both PC1 and PC2 in the principal component analysis (PCA), are primarily genes related to adipocyte biology (Fig. 5c and Supplementary Table 4). Notably, and in line with the adipogenic effect (Fig. 4), cells expressing PPARγ-R212Q are closer to control cells in the PCA plot compared with cells expressing PPARγ-E379K (Fig. 5c), indicating that the R212Q mutant more dramatically affects the ability of PPARγ to acutely activate target genes (Fig. 5a).

**Fig. 3 | The PPARγ-E379K mutation alters interaction with RXRα. a** Structure analysis and computational modeling the crystal structure of an PPARγ:RXRα complex (PPARγ in red; RXRα in gray) bound to DNA using the HADDOCK2.2 web server shows a complex interaction network involving PPARγ-E379 and -K382 (LBD) and RXRα-Y189 and -K175 (DBD) in the WT complex (left panel). PPARγ-E379K alters the configuration of this interface (middle panel). Double charge reversal mutations in PPARγ (E379K) and RXRα (K175E) can restore the PPARγ LBD-RXRα DBD interface through a novel electrostatic interaction (right panel) (PDB entry 3DZY). Amino acid residues involved in the PPARγ LBD-RXRα DBD interface are indicated in the stick format. The figures were generated by PyMOL Molecular Graphics System Version 1.8 (2015) provided by SBGrid[60]. **b** Spectroscopic analyses of helix 6 peptides. Left panel: $^1$H,$^{13}$C-HSQC spectra of PPARγ-WT$^{376-385}$ (red) and -PPARγ-E379K$^{376-385}$ (green) recorded in 20 mM Na$_2$HPO$_4$ /NaH$_2$PO$_4$ (pH 7.4) at 25 °C and overlaid. Signals originate from C$^α$. Middle panel: The C$^α$ secondary chemical shifts (SCSs) for both the WT peptide (red) and the E379K variant (green)[49]. In the WT peptide, Arg378−Leu381 showed consecutive positive Cα SCSs, indicating transient

helical structure (gray box). Right panel: Far-UV CD spectra of PPARγ-WT$^{376-385}$ (red) and PPARγ-E379K$^{376-385}$ (green) recorded at 25 °C in 20 mM Na$_2$HPO$_4$/NaH$_2$PO$_4$ (pH 7.4). Dashed vertical line indicated the minimum at 222 nm for α-helix structure. **c** HEK293T cells were transiently cotransfected with expression vectors encoding WT or mutant PPARγ, WT or mutant RXRα, and the *Lpl* PPRE-minimal promoter-reporter, in the absence or presence of 1 μM rosiglitazone. Activation of the reporter is expressed as fold induction over empty vector (control). Data are presented as mean values + SEM, with individual data points indicated with circles, $n = 3$ biologically independent experiments. One-way ANOVA with Tukey's multiple comparisons were used to compare cells transfected with mutant vs. WT; *$p < 0.05$. **d** Overexpression of the different PPARγ and RXRα proteins in HEK293T cells, as assessed by western blot analyses using a PPARγ- or RXRα- specific antibody. The arrows indicate PPARγ or RXRα, and the asterisk indicates an unknown non-specific band. Control, empty vector control; WT, wild-type. Three independent experiments were performed, and similar results were obtained. Source data for panel **b**–**d** are provided in the Source Data file.

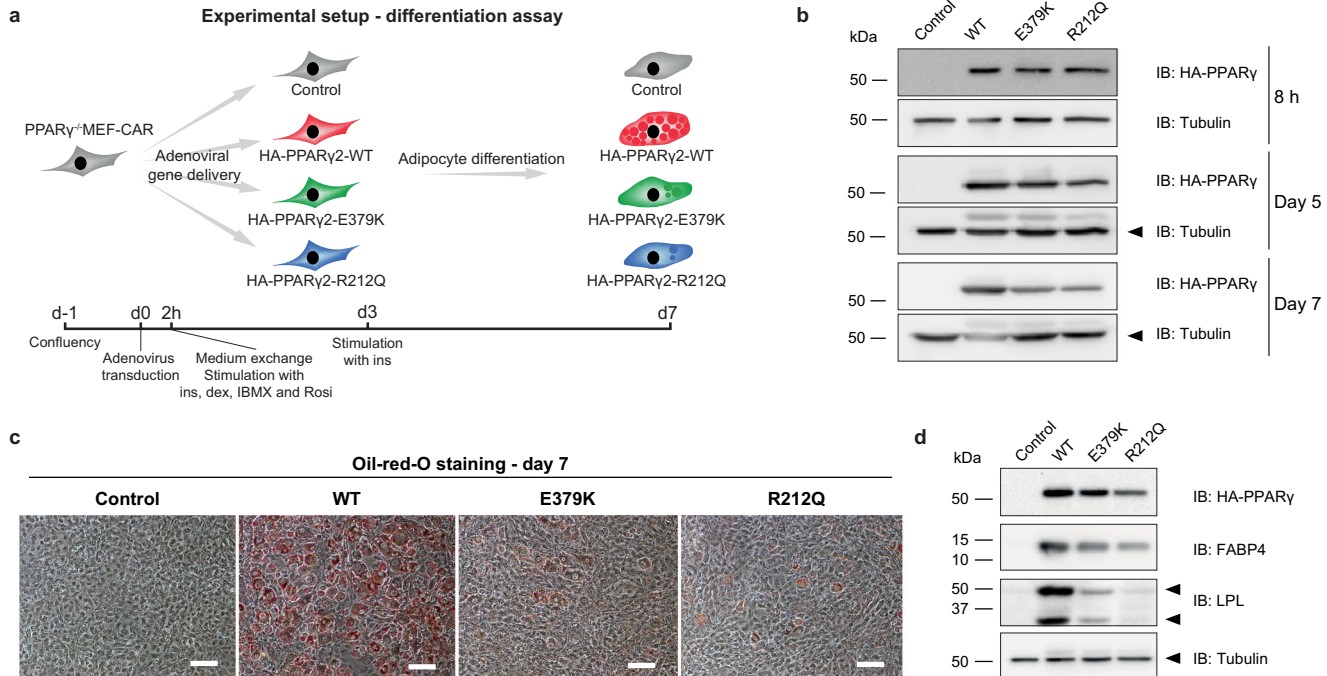

**Fig. 4 | E379K and R212Q impair the adipogenic capacity of PPARγ2.**
**a** Experimental outline showing the timing of the transduction of PPARγ$^{-/-}$ MEF-CAR cells with adenovirus containing HA-tagged PPARγ2-WT or mutants, and treatment of the transduced cells with the differentiation cocktails (2h-day 3: insulin, dexamethasone, isobutylmethylxanthine, and rosiglitazone, day 3–7: insulin). **b** Western blot assessing the expression of WT and mutant PPARγ2 at the timepoints 8 h, day 5 and day 7 after adenoviral transduction. The membrane was probed with antibodies against HA-tagged PPARγ and Tubulin (internal control). Three independent experiments were performed, and similar results were

obtained. **c** Oil-red-O staining of lipid droplets at day 7 of differentiation. Three independent experiments were performed and similar results were obtained. Scale bar, 50 μm. **d** Western blot 7 days after adenoviral transduction. The membrane was probed with antibodies against HA-tagged PPARγ, FABP4, LPL, and Tubulin (internal control). After correction for tubulin and PPARγ levels, relative protein levels for FABP4 were 77% (E379K) and 79% (R212Q) of WT levels, and for LPL 33% (E379K) and 28% (R212Q). Three independent experiments were performed and similar results were obtained. Source data for panel **b** and **d** are provided in the Source Data file.

The majority of the 277 PPARγ-WT induced genes, including the well-known PPAR target genes Krueppel-Like Factor 11 (*Klf11*)[24] and Uncoupling protein 2 (*Ucp2*)[25] were induced at comparable levels by the WT and mutant PPARγ proteins, and one gene (Pyruvate Dehydrogenase Kinase 4 (*Pdk4*)) was more induced by the mutants (Fig. 5d–f), demonstrating that mutant proteins retain some degree of functionality. Interestingly, however, 98 and 140 PPARγ-target genes are less induced by the E379K and R212Q mutant PPARγ, respectively (FDR < 5% and fold change mutant vs. WT < −1.25, Fig. 5d–f), with an overlap of 81 genes (Fig. 5g). This set of mutation-sensitive PPARγ-target genes includes several classic PPARγ-target genes like fatty acid binding protein 4 (*Fabp4*)[26], *Lpl*[18], *Cidec*[19], and CCAAT/Enhancer-

binding protein alpha (*Cebpa*)[27,28] (Fig. 5c–f and Supplementary Table 5). Notably, most genes significantly affected by only the E379K mutation showed a tendency to be affected by the R212Q mutation, and vice versa (Fig. 5h). Together, the results indicate that the PPARγ mutations E379K and R212Q, which affect different domains of the PPARγ protein and different interaction interfaces, diminish the induction of a largely overlapping subset of PPARγ-target genes, while simultaneously retaining activation potential for other target genes. Notably, mutation-sensitive target genes are generally more induced upon PPARγ-WT expression compared to target genes that are changing less than 25% comparing activation by mutant and WT PPARγ (insensitive genes) (Fig. 5i).

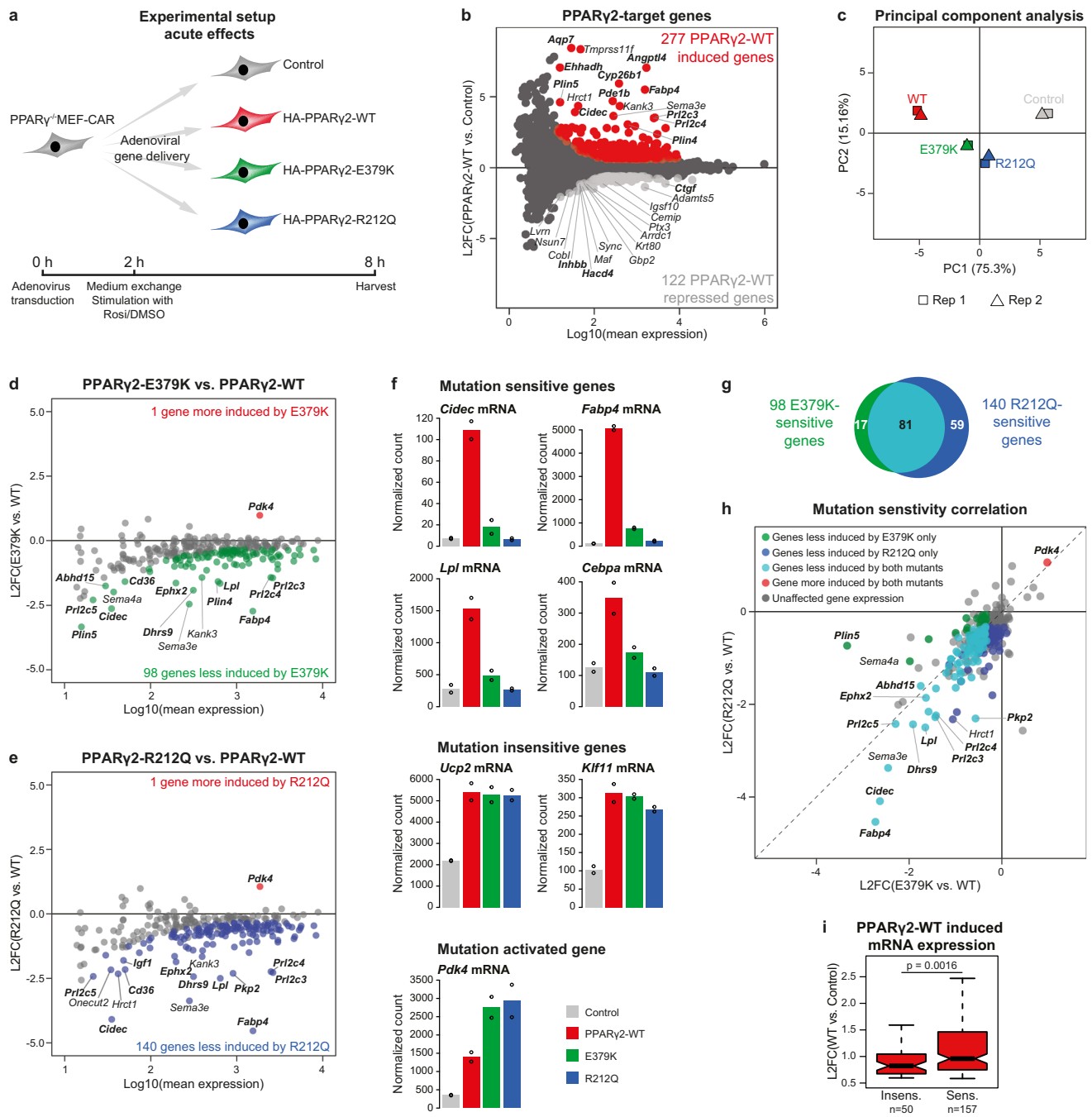

**Fig. 5 | The E379K and R212Q mutants display partial deregulation of PPARγ-target genes. a** Experimental outline showing the timing of virus transduction, ligand stimulation (1 μM Rosiglitazone) and harvest of PPARγ[-/-] MEF-CAR cells to investigate acute transcriptional changes upon introduction of WT and mutant PPARγ2s. **b** Identification of PPARγ2-target genes from RNA-seq data. PPARγ2-WT induced (red dots) and repressed (light gray dots) genes are defined using DESeq2 with Benjamini–Hochberg correction (padj. <0.05, two-sided) and increasing or decreasing by a fold change >1.5 compared to control cells, respectively. Black dots represent unaffected genes. L2FC, log2 fold change. Top-15 most induced and repressed genes are indicated, with genes known to be involved in adipocyte biology marked in bold. **c** Variance in RNA-seq data (n = 2 independent biological experiments). Identification of **d** E379K-sensitive and **e** R212Q-sensitive PPARγ2-target genes. Red dots represent genes more induced by mutant compared to WT PPARγ2 (padj.(mut vs. WT) < 0.05, FC (mut vs. WT) > 1.25). Green and blue dots represent genes less induced by E379K and R212Q compared to WT, respectively (padj.(mut vs. WT) < 0.05, FC (mut vs. WT) < −1.25). Statistical significance was determined by DESeq2 using Benjamini–Hochberg correction, two-sided test. L2FC, log2 fold change. Top-15 less induced genes are indicated. Genes known to be involved in adipocyte biology are marked in bold. **f** Bar plots indicating RNA-seq based expression of selected PPARγ2-target genes. Bars represents mean of independent biological replicates (n = 2), dots indicate individual replicates. **g** Venn diagram representing overlap of genes that are significantly downregulated by PPARγ2-E379K and -R212Q. **h** Correlation of sensitivity to the E379K and R212Q mutations relative to WT (L2FC, log2 fold change). Top-10 most affected genes for each mutant vs. WT is indicated. **i** Boxplot displaying induction levels for genes affected by one or both mutations (sens., n = 157) or genes changing less than 25% (mutant vs. WT) (insens., n = 50). L2FC, log2 fold change. Data are presented as notch, median; box, first and third quartiles; whiskers, 1.5 times the interquartile range. Statistical significance was determined by two-sided unpaired two-samples Wilcoxon–Mann–Whitney test. Source data for panel **b**–**i** are provided in the Source Data file.

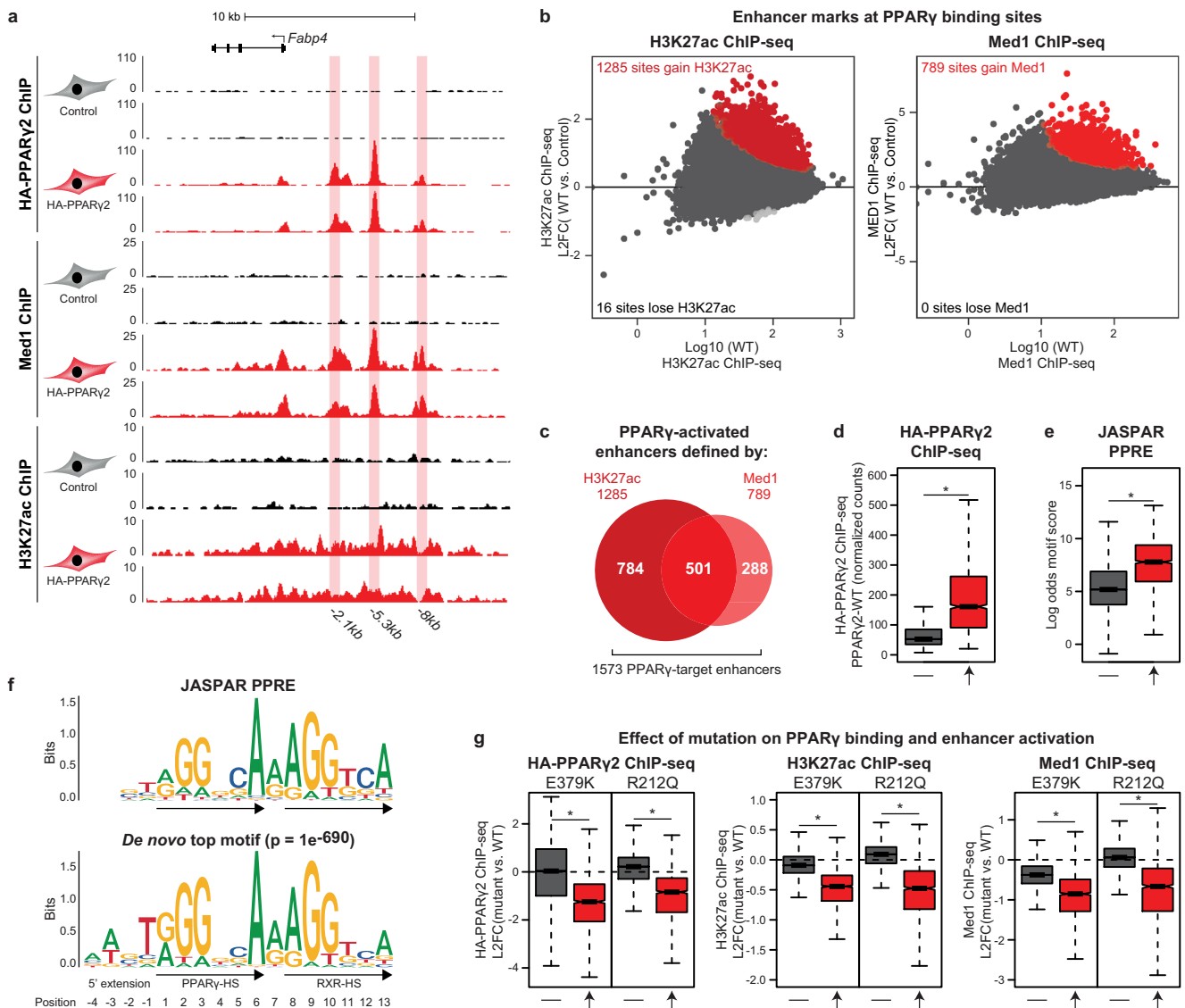

**Fig. 6 | A subset of PPARγ-bound sites are functional PPARγ-target enhancers.**
**a** UCSC Genome Browser screenshot showing HA-PPARγ, Med1, and H3K27ac ChIP-seq in control and PPARγ2-WT expressing cells at the *Fabp4*-locus. PPARγ-binding sites (identified using Homer findPeaks and extended to 500 bp and passing a cutoff of <35 tags) are highlighted. Annotation of PPARγ-binding sites are relative to the transcriptional start site (TSS) of *Fabp4*. **b** Identification of enhancers activated by PPARγ2-WT. H3K27ac and Med1 ChIP-seq signal was counted within 41830 PPARγ binding sites extended ±1500 bp (H3K27ac) or ±250 bp (Med1) from peak center. Red and light gray points indicate enhancers that gain or lose ChIP-seq signal, respectively, upon PPARγ2-WT expression. Significance was determined by DESeq2 with Benjamini–Hochberg correction, two-sided test(FDR < 0.1). L2FC, log2 fold change. **c** Venn diagram showing the number of activated enhancers defined by gain in H3K27ac and/or Med1 ChIP-seq signal, in total identifying 1573 PPARγ-target enhancers. **d** Boxplot of PPARγ ChIP-seq signal in enhancers that does not gain or lose enhancer activity upon PPARγ2-WT expression (−, n = 40241), or at PPARγ-target enhancers as defined in panel **b**, **c** (↑, n = 1573). **e** JASPAR PPRE-motif score of the highest scoring motif within ±100bp from peak center. −: Non-activated enhancers, ↑: PPARγ-target enhancers. **f** Position weight matrix (PWM) for the JASPAR PPRE (top) and the de novo top motif (bottom). De novo motif search was made using Homer findMotifsGenome and searched within ±100 bp of peak center of PPARγ-target enhancers with motif length of 15–17 bases. PPAR-HS PPAR-half site, RXR-HS RXR-half site. **g** Mutations affect primarily PPARγ-target enhancers. Boxplots showing the log2 fold change (L2FC) (mutant vs. WT) for HA-PPARγ, H3K27ac, and Med1 ChIP-seq signal in non-activated (−) and activated (↑) enhancers. For all boxplots, data are presented as notch, median; box, first and third quartiles; whiskers, 1.5 times the interquartile range. *p < 2e-16 using two-sided unpaired two-samples Wilcoxon–Mann–Whitney test. Source data for panel **b**, and **d**–**g** are provided in the Source Data file.

## E379K and R212Q mutants affect recruitment of PPARγ to a subset of PPARγ-target enhancers

The finding that R212Q and E379K mutant proteins compromise transactivation of only a subset of target genes and that this subset is mostly shared between the mutants is intriguing and indicates that a subcategory of enhancers may be particularly sensitive to PPARγ mutations.

HA-PPARγ chromatin immunoprecipitation sequencing (ChIP-seq) of PPARγ⁻/⁻ MEF-CAR cells transduced with adenoviral vectors expressing PPARγ for 2 h followed by treatment with rosiglitazone for

6 h revealed 41,830 PPARγ-WT binding sites (four-fold enrichment above local background, p < 0.0001) (Fig. 6a). To evaluate the importance of PPARγ at these binding sites, we performed H3K27 acetylation (H3K27Ac) and Med1 ChIP-seq, which are well-established proxies for enhancer activity. Notably, for the majority of PPARγ binding sites, PPARγ recruitment is not associated with significant changes in H3K27Ac or Med1 occupancy within the time frame of the experiment (Fig. 6b). Thus, PPARγ recruitment leads to increased H3K27Ac at only 1285 sites and MED1 recruitment at 789 PPARγ binding sites (FDR < 0.1), (Fig. 6b), with 501 sites displaying an increase in both marks

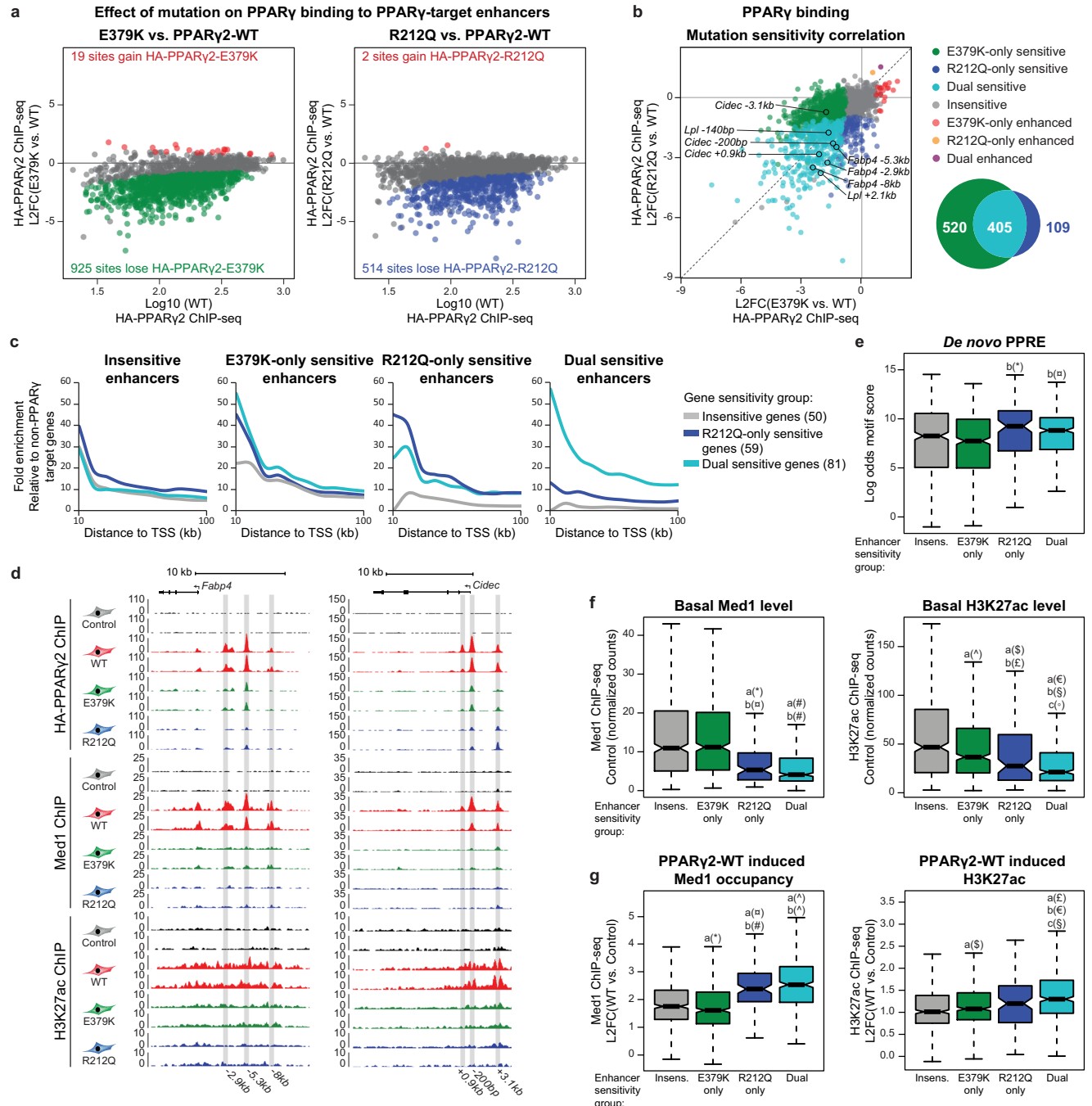

**Fig. 7 | PPARγ target enhancers display different sensitivity to mutations dependent on basal enhancer activity. a** MA-plots of PPARγ2-E379K (left) and PPARγ2-R212Q (right) binding relative to PPARγ2-WT binding (L2FC, log2 fold change). Gained and lost binding is defined by padj. (mut vs. WT) < 0.05 (DESeq2 with Benjamini–Hochberg correction, two-sided) and the binding intensity (mut vs. WT) increasing or decreasing by at least 25%, respectively. **b** Correlation of sensitivity to E379K mutation and R212Q mutation (left) (L2FC, log2 fold change). Selected enhancers are indicated. Venn-diagram of enhancers with mutation-facilitated reduced PPARγ-binding (right). **c** Enrichment of enhancers in the vicinity of PPARγ-target genes relative to the number of enhancers in the vicinity of 200 randomly selected genes not regulated by PPARγ. R212Q-only and dual-sensitive genes are defined as in Fig. 5h. Insensitive target genes are changing less than 25% comparing mutant vs. WT PPARγ. E379K-only sensitive genes are excluded from the analysis as the gene group is very small (17 genes). (TSS, transcriptional start site). **d** UCSC Genome Browser track showing HA-PPARγ, Med1, and H3K27ac ChIP-seq at the *Fabp4*-locus (left) and *Cidec*- locus (right). PPARγ-target enhancers are

highlighted. **e** Boxplot of log odds motif score for the de novo PPRE within groups of enhancers. Significance was assessed by two-sided pairwise Wilcoxon rank sum tests with Benjamini–Hochberg correction, a, versus insensitive enhancer group (insens., $n = 519$); b, versus E379K-only sensitive enhancer group ($n = 520$); c, versus R212Q-only sensitive enhancer group ($n = 109$); dual-sensitive group ($n = 405$), *$p = 0.0005$, ¤$p = 5.7e$-5. **f** Boxplot of basal Med1 (left) and H3K27ac (right) ChIP-seq signal within enhancer groups. Significance was assessed as in panel **e**. *$p = 4.8e$-10, ¤$p = 3.3e$-10, #$p < 2e$-16, ˆ$p = 0.0030$, \$$p = 0.00062$, £$p = 0.024$, €$p < 2e$-16, §$p = 2.4e$-14, °$p = 0.046$. **g** Boxplot of PPARγ2-WT induced changes in Med1 and H3K27ac ChIP-seq signal within enhancer groups (L2FC, log2 fold change). Significance was assessed as in panel **e**. *$p = 0.0037$, ¤$p = 1.4e$-11, #$p = 4.3e$-16, ˇ$p < 2e$-16, \$$p = 0.024$, £$p = 9.4e$-12, €$p = 1.1e$-7, §$p = 0.049$. Data in boxplots are presented as notch, median; box, first and third quartiles; whiskers, 1.5 times the interquartile range. Source data for panel **a–c** and **e–f** are provided as a Source Data file.

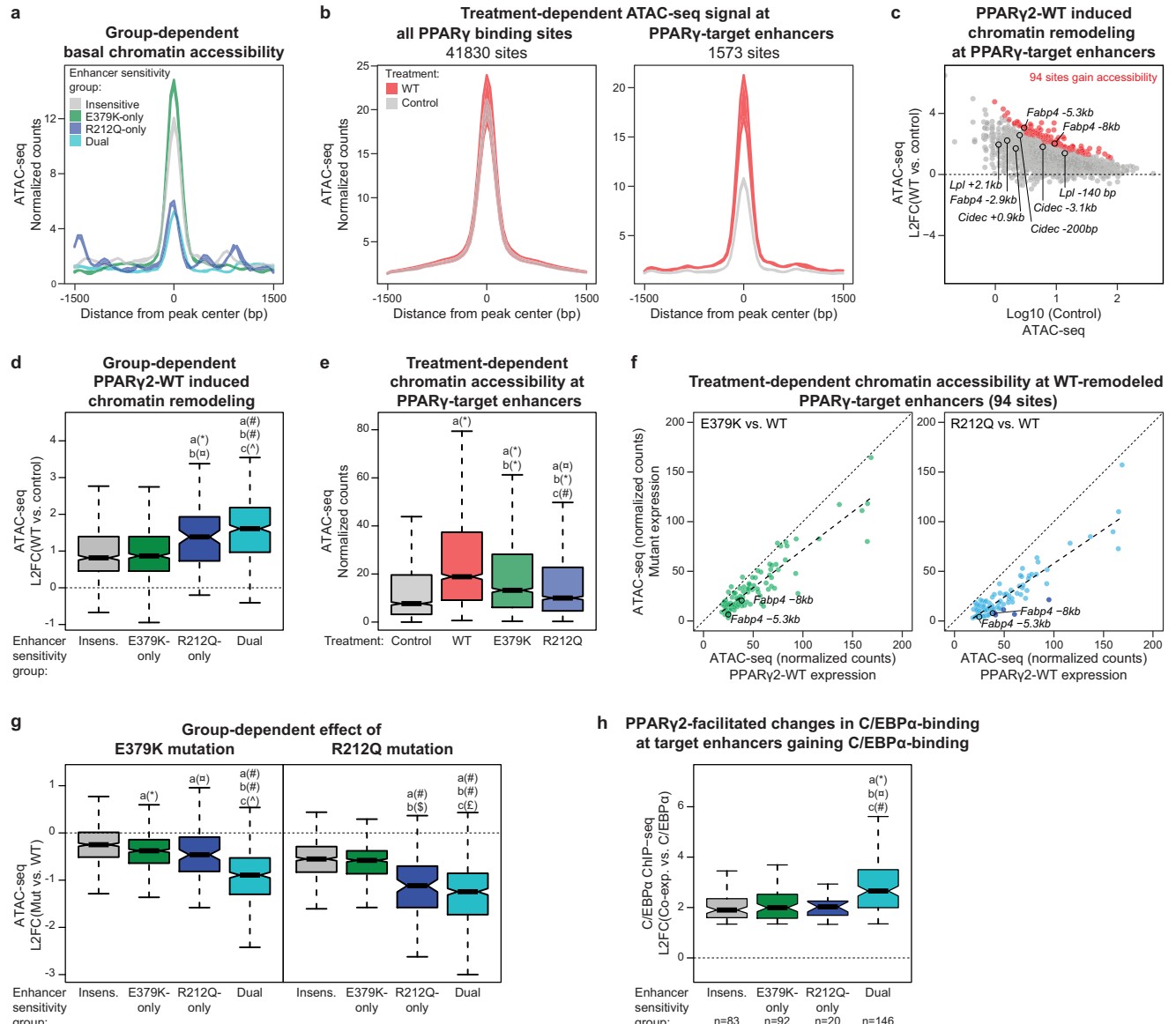

**Fig. 8 | R212Q-sensitive enhancers are found in less accessible regions of the chromatin. a** Enhancer group-dependent ATAC-seq signal in control cells. Colored area represents the difference between the two replicates. **b** Treatment-dependent ATAC-seq signal from control and PPARγ2-WT expressing PPARγ⁻/⁻ MEF-CAR cells on all PPARγ binding sites (left) or PPARγ-target enhancers (right). Colored area represents the difference between the two replicates. **c** MA-plot of PPARγ2-WT induced chromatin accessibility at PPARγ-target enhancers (L2FC, log2 fold change). Gained accessibility is defined using DESeq2 with Benjamini–Hochberg correction, two-sided test, padj <0.05, L2FC(WT vs. control) > 0. Selected enhancers are indicated. **d** Enhancer group-dependent PPARγ2-WT induced chromatin remodeling. Significance was assessed by two-sided pairwise Wilcoxon rank sum tests with Benjamini–Hochberg correction, a, versus insensitive enhancer group (n = 519); b, versus E379K-only sensitive enhancer group (n = 520); c, versus R212Q-only sensitive enhancer group (n = 109), dual-sensitive enhancer group (n = 405), *p = 1.7e-7, ¤p = 4.7e-7, #p < 2e-16, ^p = 0.01. **e** Treatment-dependent ATAC-seq signal at PPARγ-target enhancers (n = 1573). Significance was assessed by two-sided

pairwise Wilcoxon rank sum tests with Benjamini–Hochberg correction, a, versus control; b, versus WT expression; c, versus E379K-expression, *p < 2e-16, ¤p = 2.3e-9, #p = 6.8e-9. **f** ATAC-seq signal of E379K (left) and R212Q (right) treated cells relative to the WT ATAC-seq signal at enhancers where PPARγ2-WT significantly induces remodeling (94 sites defined in panel **c**). Linear models fitting the data are indicated with dashed lines. Enhancers that significantly lose ATAC-seq signal upon R212Q mutation are marked with dark blue (DESeq2 with Benjamini–Hochberg correction, two-sided test, padj <0.05, L2FC(R212Q vs. WT) < 0). Selected enhancers are indicated. **g** Mutation-induced changes in remodeling capacity within enhancer groups. Significance was assessed as in panel **d**. *p = 1.2e-6, ¤p = 0.00027, #p < 2e-16, ˇp = 1.9e-11, $p = 8e-16, £p = 0.043. **h** Enhancer group-dependend PPARγ-induced gain in C/EBPα ChIP-seq signal upon PPARγ2 and C/EBPα co-expression in PPARγ⁻/⁻ MEF-CAR cells. Data from Madsen et al.[6]. Significance was assessed as in panel **d**. *p = 2.2e-6, ¤p = 1.7e-6, #p = 0.0014. Data in boxplots are presented as notch, median; box, first and third quartiles; whiskers, 1.5 times the interquartile range. Source data are provided in the Source Data file.

(Fig. 6c). Based on this, we defined 'PPARγ-target enhancers' as the 1573 putative enhancers that gain H3K27ac and/or Med1 in response to PPARγ binding. Importantly, PPARγ-target enhancers display higher average PPARγ occupancy and a stronger PPRE motif, compared with the other binding sites (Fig. 6d, e), and the top-scoring de novo motif found within PPARγ-target enhancers resembles the JASPAR PPRE (Fig. 6f). Taken together, this indicates that PPARγ-activated sites are

functional PPARγ binding sites, where PPARγ acts as a key driver of enhancer activity.

To investigate how the E379K and R212Q mutations affect the ability of PPARγ to bind to chromatin and activate enhancers, we repeated the PPARγ, H3K27ac, and MED1 ChIP-seq for the PPARγ mutants. Interestingly, the mutations did not affect overall PPARγ association with chromatin but selectively compromised PPARγ

binding and enhancer activation at PPARγ-target enhancers (Fig. 6g). Taken together, these results indicate that only a small subset of PPARγ binding sites are functionally important PPARγ-target enhancers, and that PPARγ recruitment to these on average is sensitive to mutations that destabilize the PPARγ:RXR heterodimer binding to DNA.

To better understand how the two mutations affect the recruitment of PPARγ to target enhancers, we analyzed how mutations affect the binding of PPARγ to each of the PPARγ-target enhancers. Out of the 1573 target enhancers (Fig. 6c), E379K decreased binding to 925 enhancers, while R212Q decreased binding to 514 enhancers (Fig. 7a). Interestingly, the reduced ability of the mutant PPARγ proteins to engage with chromatin defines clear subclasses of enhancers, where 405 enhancers display reduced binding of both mutants, whereas 520 and 109 enhancers are selectively less bound by E379K and R212Q, respectively (Fig. 7b). Dual-sensitive enhancers are highly enriched in the vicinity of dual and R212Q-only sensitive genes compared to insensitive genes, and R212Q-only sensitive enhancers are more often found in the vicinity of dual and R212Q-only sensitive genes (Fig. 7c). These findings indicate that the decreased binding of the mutants to target enhancers is functionally linked to their reduced activation potential. For instance, binding of PPARγ to the well-known PPREs controlling expression of the mutation-sensitive genes Fabp4[29], Cidec[19], and Lpl[18] is compromised by the mutations, while binding of PPARγ to the Ucp3 intronic PPRE controlling expression of the mutation-insensitive gene Ucp2[25] is unaffected by the mutations (Fig. 7d and Supplementary Fig. 6).

Functional characterization of the different subclasses of enhancers showed that enhancers that primarily are sensitive to R212Q (R212Q-only) are characterized by strong PPRE motifs (Fig. 7e), low levels of enhancer activity in the non-transduced cells (Fig. 7f) and a high fold increase in enhancer activity (Med1 recruitment) in response to PPARγ-WT (Fig. 7g). This indicates that these represent enhancers that are highly dependent on PPARγ for activation. In contrast, enhancers primarily sensitive to E379K are characterized by a weaker PPRE (Fig. 7e), high levels of enhancer activity in the non-transduced cells (Fig. 7f), and less fold increase in Med1 recruitment in response to PPARγ-WT, indicating that these are enhancers where PPARγ plays a more modest role. The enhancers that are sensitive to both mutations generally share many characteristics of the R212Q-only enhancers, including strong PPRE motifs (Fig. 7e) and dependency on PPARγ for activation (Fig. 7f, g). Finally, PPARγ-target enhancers that are insensitive to mutations are more similar to E379K by having high levels of enhancer activity in the absence of PPARγ (Fig. 7f) and by being more modestly activated by PPARγ (Fig. 7g). Consistent with that, receiver-operating characteristic (ROC) analysis indicates that activity of the PPARγ-target enhancer in the non-transduced cells is the best predictor of mutation sensitivity (Supplementary Fig. 7).

Taken together, PPARγ-target enhancers with low activity in the absence of PPARγ are, on average sensitive to both mutations, especially the R212Q mutation, whereas PPARγ-target enhancers, which are already active prior to expression of PPARγ are primarily sensitive to the E379K mutation.

### E379K and R212Q mutants decrease the remodeling capacity of PPARγ

Local chromatin structure constitutes a major determinant of the ability of transcription factors to bind to DNA[30], and we hypothesized that chromatin accessibility could discriminate between mutation-sensitive and insensitive-sites. We therefore determined chromatin accessibility in non-transduced PPARγ$^{-/-}$ MEF-CARs using an assay for transposase-accessible chromatin sequencing (ATAC-seq). Notably, examination of a 3 kb window centered around the PPARγ peak shows that R212Q-only and dual-sensitive PPARγ-target enhancers have low chromatin accessibility (i.e., are nucleosome-rich) in chromatin prior to expression of PPARγ, whereas insensitive and E379K-only sensitive

enhancers are already accessible in non-transduced cells (Fig. 8a and Supplementary Fig. 8). These findings are consistent with the notion that the R212Q-sensitive enhancers are inactive in the non-transduced state and highly dependent on PPARγ for activation.

To examine the ability of PPARγ to drive the remodeling of target enhancers, we assessed chromatin accessibility before and after transduction with Ad-mPPARγ2-WT. Whereas PPARγ expression does not affect average chromatin accessibility at all PPARγ binding sites, it specifically increases chromatin accessibility at PPARγ-target enhancers, with 94 enhancers significantly gaining accessibility (FDR < 0.05) by ATAC-seq (Fig. 8b, c). Notably, R212Q-sensitive enhancers gain more accessibility than insensitive and E379K-only enhancers (Fig. 8d), consistent with the higher gain in activity at these enhancers (Fig. 7g).

The finding that R212Q-sensitive enhancers are relatively inaccessible and inactive prior to expression of PPARγ indicates that the R212 residue, possibly through its direct interaction with the minor groove in DNA[11–13], is important for the ability of PPARγ to bind to and remodel inactive PPARγ-target enhancers. To directly assess this, we compared the ability of PPARγ-WT and the R212Q and E379K mutants to remodel target enhancers. While both mutations significantly diminish PPARγ-induced remodeling, the R212Q mutation has the most dramatic effect (Fig. 8e, f). Intriguingly, however, the R212Q mutation also leads to slightly decreased remodeling and activation of target enhancers, where PPARγ binding is not affected by the mutation (insensitive and E379K-only enhancers) (Fig. 8g and Supplementary Fig. 9). Taken together, these findings indicate that the R212Q mutation in PPARγ compromises the ability to bind to enhancers in closed chromatin and induce the remodeling and activation of these enhancers, but that the mutation also affects the activation of enhancers independent of the level of PPARγ binding. Similarly, the E379K mutation affects the remodeling and activation of R212Q-only enhancers (Fig. 8g and Supplementary Fig. 9), suggesting that this mutation also modestly affects the activation of enhancers independent of PPARγ binding.

We have previously shown that the two adipocyte master regulators, PPARγ and C/EBPα, can potentiate the binding of each other to shared binding sites in closed chromatin in PPARγ$^{-/-}$ MEF-CAR cells[6]. Of interest to this work, we showed that the ability of PPARγ2 to act as a leading factor facilitating C/EBPα binding appeared to be dependent on a strong PPRE, indicating that PPARγ requires a strong PPRE to bind to DNA in nucleosome-embedded chromatin. Interestingly, by comparing these data with the current study, we found that dual-sensitive PPARγ target enhancers, which are 1) most inactive in the absence of PPARγ (Fig. 7f); 2) most sensitive to PPARγ-induced remodeling (Fig. 8d); and 3) most sensitive to PPARγ mutations (Fig. 8g), generally are enhancers where PPARγ act as the leading factor for C/EBPα (Fig. 8h).

### R212Q-sensitive enhancers contain PPREs with consensus 5′-extension

Since our data indicated that R212 in PPARγ is important for the recruitment of PPARγ to closed chromatin, we dissected the PPRE motif into the 5′ extension, the PPARγ half site, and the RXR-half site (Fig. 6f) and assessed the motif score within each of these segments for the different subgroups of PPARγ-target enhancers (Fig. 9a). Interestingly, the 5′ extension, as well as the PPARγ half site, has a higher motif score in R212Q-sensitive enhancers compared to E379K-only and insensitive target enhancers.

To further investigate the importance of the PPRE-motif strength, we classified PPARγ-target enhancers as open or closed based on their basal accessibility and assessed the effect of mutations on PPARγ binding for different intervals of PPRE-motif score (Fig. 9b). This shows that when the target enhancer is in accessible chromatin, WT PPARγ and both E379K and R212Q mutants are recruited equally well to PPREs with high motif strength (>12),

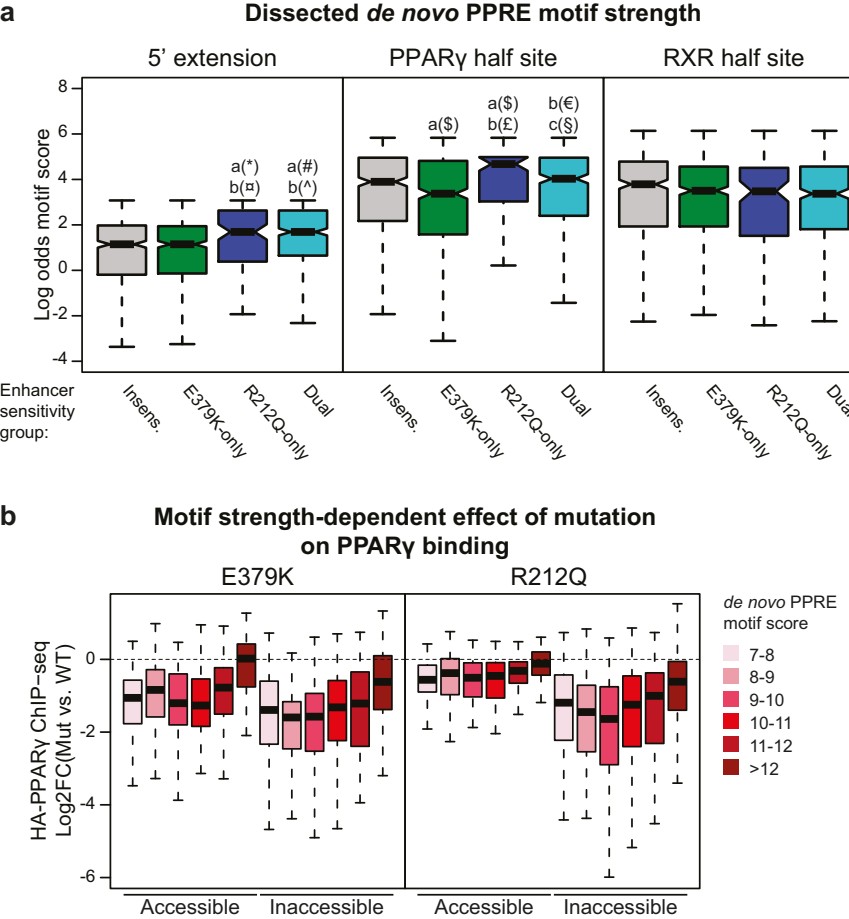

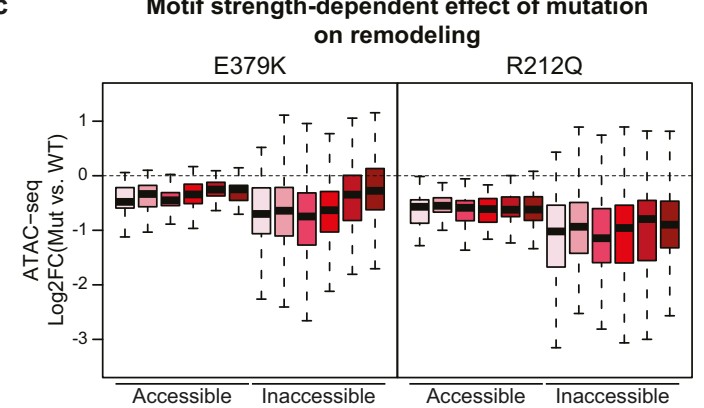

**Fig. 9 | R212Q-sensitive enhancers are characterized by a stronger 5'-extension and PPARγ-half site. a** The de novo PPRE (Fig. 6f) was dissected into the 5' extension, PPARγ-half site, and RXR-half site and the motif strength was assessed within these sections for each group of enhancers. Significance was assessed by two-sided pairwise Wilcoxon rank sum tests with Benjamini–Hochberg correction, a, versus insensitive enhancer group (n = 519); b, versus E379K-only sensitive enhancer group (n = 520); c, versus R212Q-only sensitive enhancer group (n = 109), dual-sensitive enhancer group (n = 405), *p = 0.00022, ¤p = 5.8e-5, #p = 5.7e-10, ^p = 8.1e-12, $p = 0.0052, £p = 2e-5, €p = 0.0037, §p = 0.011. Boxplots are presented as notch, median; box, first and third quartiles; whiskers, 1.5 times the interquartile range.

**b**, **c** PPARγ-target enhancers were divided into accessible (normalized counts > 15 tags) and inaccessible (normalized counts <15 tags) enhancers based on the ATAC-seq data. The effect of mutation on **b** PPARγ binding and **c** ATAC-seq signal was assayed dependent on the PPRE-motif strength within the PPARγ-target enhancer. Accessible enhancers: $n_{\text{Motif score 7-8}} = 53$; $n_{\text{Motif score 8-9}} = 53$; $n_{\text{Motif score 9-10}} = 63$; $n_{\text{Motif score 10-11}} = 55$; $n_{\text{Motif score 11-12}} = 26$; $n_{\text{Motif score >12}} = 27$. Inaccessible enhancers: $n_{\text{Motif score 7-8}} = 92$; $n_{\text{Motif score 8-9}} = 129$; $n_{\text{Motif score 9-10}} = 142$; $n_{\text{Motif score 10-11}} = 176$; $n_{\text{Motif score 11-12}} = 92$; $n_{\text{Motif score >12}} = 95$. Boxplots are presented as bold line, median; box, first and third quartiles; whiskers, 1.5 times the interquartile range. Source data are provided in the Source Data file.

whereas recruitment is negatively affected by the E379K mutation but not the R212Q mutation when the motif score is decreasing. Importantly, for target enhancers in inaccessible chromatin, the ability of both mutants to bind to target enhancers is highly dependent on the motif strength.

Taken together, these results show that the R212 residue is important for the ability of PPARγ to recruit to sites in closed chromatin, whereas the binding to nucleosome-free accessible target sites is much less affected by this mutation. Interestingly, however, the remodeling capacity of PPARγ-R212Q is compromised independently

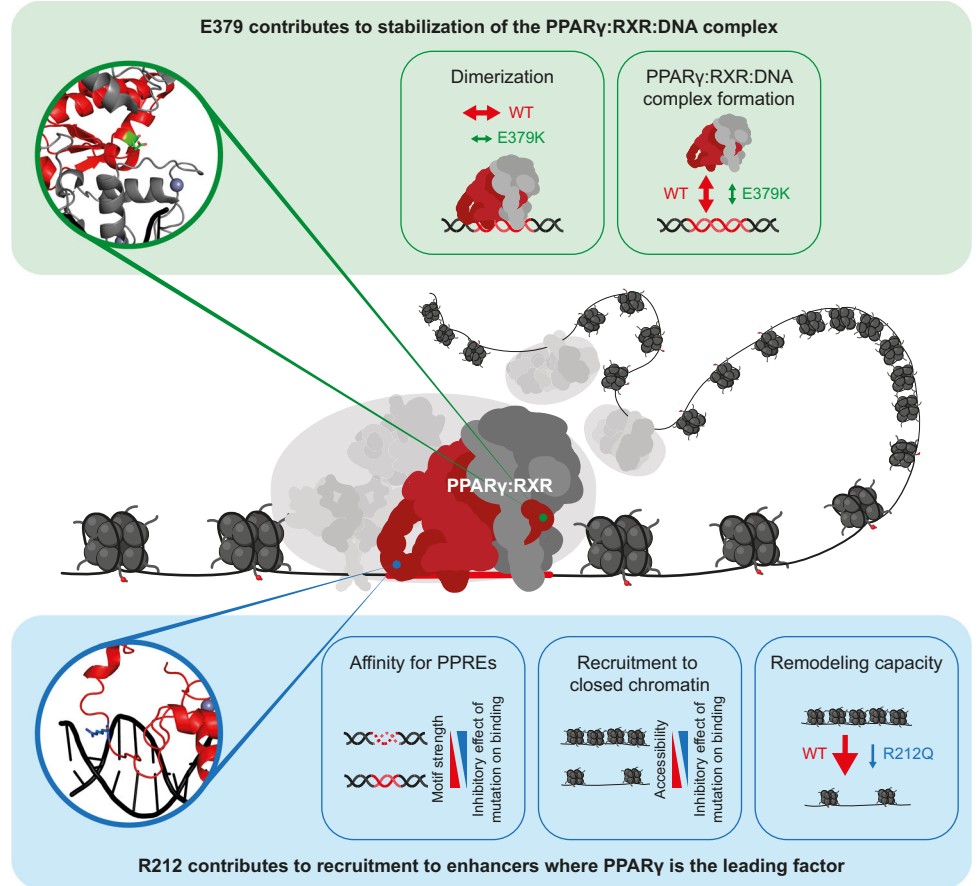

**Fig. 10 | Model illustrating how E379K and R212Q lipodystrophy mutants reveal important interaction interfaces in the PPARγ:RXR:DNA complex.** Central panel: PPARγ:RXR:DNA ternary complex with the respective positions of E379 and R212 residues based on the crystal structure[11]. Top panel: the molecular effect of the E379K mutation. Lower panel; the molecular effect of the R212Q mutant. See text for further explanation.

of motif score and PPARγ recruitment, suggesting that PPARγ-R212 is particularly important for the ability of PPARγ to remodel chromatin (Fig. 9c). The E379K mutation also displays the greatest effect on PPARγ binding to inaccessible sites; however, in addition, it decreases binding of PPARγ to accessible target sites in a motif-strength-dependent manner. This suggests that the motif becomes more important for the formation of the ternary PPARγ:RXR:DNA complex when the PPARγ:RXR heterodimer is destabilized by the E379K mutation. Collectively, these data reveal novel mechanisms contributing to the stabilization of the binding of the PPARγ:RXR dimer to DNA, which is particularly important in enhancers where PPARγ acts as the leading factor.

## Discussion

Structure-based studies have indicated extensive intermolecular interactions in the DNA-bound PPARγ:RXR heterodimer[11–13], but how these interactions translate into gene activation in a cellular context, i.e. in cooperation with other transcription factors and in the context of chromatin, is currently poorly understood. Here we employed two previously uncharacterized FPLD3-associated PPARγ mutants, predicted to be involved in intermolecular interactions in the PPARγ:RXR:DNA complex, to probe the molecular mechanisms underlying enhancer activation by PPARγ. We show that the R212Q and E379K mutations in PPARγ lead to reduced adipogenic potential and impaired activation of a subset of PPARγ target genes. Consistent with this, genome-wide analyses of the transcriptional and epigenomic properties of the mutants demonstrate that the mutations only affect the activation of a subset of PPARγ-target enhancers and that a major determinant of sensitivity to these mutations is low chromatin

accessibility. Interestingly, we show that the two mutations affect enhancer activity by distinct mechanisms, thereby providing insight into how PPARγ engages with its target sites. Furthermore, as important intra- and intermolecular interactions have been reported in other nuclear receptors (e.g. RARβ[9] and GR[31]), it will be of interest to investigate whether disease-causing mutations in nuclear receptors other than PPARγ have similar effects[32].

Based on the crystal structure, the R212Q mutation is expected to decrease the ability of the PPARγ hinge to engage in interactions within the minor groove of the DNA helix immediately 5′ of the PPRE (referred to as the 5′ extension or 5′ upstream region; 5′ UR)[11]. Previous studies based on selected target sites indicated that the 5′extension is required for PPAR:RXR binding to suboptimal direct repeats and that this sequence provides specificity of the PPRE towards the PPAR:RXR heterodimer[33,34]. Consistent with a role for the interaction between the PPARγ hinge and the 5′ extension of the PPRE in chromatin, genome-wide profiling of PPARγ occupancy by ChIP-seq has shown that the 5′ extension is enriched in natural PPARγ:RXR binding sites[3,4,6]. Here we show that the R212Q mutation interferes with the ability of PPARγ to recruit to and remodel target sites in nucleosome-rich chromatin, whereas there is little effect on binding to accessible nucleosome-free target sites. This indicates that the interaction between the PPARγ hinge and the 5′extension of the PPRE is particularly important for the ability of PPARγ to bind to target sites in nucleosome-rich chromatin. In addition, this interaction appears to enhance chromatin remodeling capacity independent of PPARγ binding, possibly by increasing intermolecular interactions in the PPAR:RXR:co-factor complex, or by interfering with the interaction between DNA and nucleosomes.

According to crystallography and SAXS studies, the E379 residue is located in the LBD at a position that contacts the DBD of RXRα in the DNA-bound heterodimer[11,14]. The existence of the PPARγ LBD-RXRα DBD heterodimerization interface is a topic of debate as this interface could not be observed in solution by SAXS[12,13]. However, our double charge reversal mutation assay, combining the E379K mutant with RXRα-K175E on the *Lpl −140bp* PPRE, suggests that this interface contributes to PPARγ-mediated transcriptional activity. Interestingly, another recently identified variant in the *PPARG* locus giving rise to a S383R mutant PPARγ[35] is predicted to affect the same interaction interface; however, no functional data are available yet. Like the R212Q mutation, the E379K mutation has the greatest impact on the binding of PPARγ to inaccessible target sites in chromatin. However, this mutation also affects binding to accessible sites, indicating that this mutation leads to a more general destabilization of the PPARγ:RXR heterodimer. This is consistent with the observation that the E379K mutation affects PPARγ binding to more target enhancers than the R212Q mutation.

The finding that the R212Q mutation has the greatest effect on enhancer activity and target gene expression compared with the E379K mutation despite affecting PPARγ recruitment to fewer enhancers is intriguing. Most likely, this is due to the fact that the R212Q mutation leads to a more dramatic loss of remodeling capacity at target sites, including dual-sensitive target sites. Thus, whereas the E379K mutation affects PPARγ binding to many enhancers through destabilization of the PPARγ:RXR heterodimer, the R212Q mutation more severely compromises the ability of PPARγ to act as a facilitating transcription factor driving remodeling (Fig. 10). Since the need for PPARγ to act as a leading transcription factor is highly dependent on which other transcription factors are expressed, one would expect that the phenotypic result of the R212Q mutant is highly context-dependent.

Taken together, our comprehensive investigations of the functional and biochemical properties of two PPARγ FPLD3-associated mutants highlight the importance of intermolecular interactions for transcriptional activation. Our data indicate that the interaction between the hinge region of PPARγ and the 5' extension of the PPRE is particularly important for the ability of PPARγ to bind to and remodel enhancers in inaccessible chromatin. Furthermore, our data support the existence of a functional interface between the RXR-DBD and helix 6 in the LBD of PPARγ and indicate that this interaction plays a role in the general stabilization of the PPARγ:RXR:DNA ternary complex. Overall, this work underscores the importance of intermolecular interactions in PPARγ functions and indicates that the subtle molecular defects in these interactions are sufficient to cause FPLD3.

## Methods

The study was conducted according to the 1964 declaration of Helsinki and its later amendments or compatible ethical standards. The study was approved by the Ethics Committee of University Hospitals Leuven (File S57866) and the Ethics Committee NedMec (File 22–891). Written informed consent for study participation and publication was obtained from the index patients.

### Index subjects and DNA sequence analysis

Both index patient 1 and 2 displayed typical features of partial lipodystrophy: a clear excess of subcutaneous fat on the face, neck, trunk, and abdomen with a lack of subcutaneous fat on the extremities. Additional clinical features are indicated in Fig. 1a, b. Genomic DNA was extracted from peripheral-blood leukocytes in venous blood samples using QIAamp DNA Blood Mini Kit according to the manufacturer's instructions (Qiagen Hilden, Germany). We sequenced all coding exons of *PPARG, AKT2, CIDEC*, and *LMNA* (exon 8–9), including intron-exon boundaries of the index patients. Primers are available upon request. In index patient 1 we identified a heterozygous mutation in exon 6 of the *PPARG* gene corresponding to c.1135 G > A and p.E379K in reference sequences NM_015869 and NP_056953.2, respectively. This variant is absent in the variation databases NHLBI GO Exome sequencing Project (ESP) (Exome Variant Server), 1000Genomes (Consortium, 2015), Single Nucleotide Polymorphism Database (dbSNP)[36], and The Genome Aggregation Database (gnomAD)[37]. Amplification and Sanger sequencing of exon 6 and flanking regions were performed to genotype family members. In index patient 2, we identified a heterozygous mutation in exon 5 of *PPARG* corresponding to c.635 G > A and p.R212Q in reference sequences NM_015869 and NP_056953.2, respectively. Genotype data from family members were not available. The same variant was previously reported independently in a female subject with FPLD3 but not functionally characterized[16].

### Cell culture

The murine PPARγ$^{-/-}$ MEF-CAR cell line[6], human osteosarcoma cell line U2OS, and the human embryonic kidney cell line HEK293T were maintained in Dulbecco's modified Eagle's medium (DMEM; Gibco) 4.5 g/L D-glucose supplemented with 10% fetal bovine serum (Gibco), and 100 µg penicillin/ml and 100 µg streptomycin/ml (Invitrogen).

### Luciferase-based reporter assays

The pGL3-*mLpl*-PPRE-Luc2 (lipoprotein lipase; *Lpl*) reporter construct was generated by the insertion of one copy of the mouse *Lpl* −160 bp PPRE[18] upstream of the minimal promoter and one copy downstream of the luciferase gene in a pGL3-minimal promoter-luciferase 2 (Luc2) backbone (Supplementary Fig. 2)[38]. The pGL3-*mLpl*-PPRE-Luc2 was used as a template to generate Lpl 5' upstream region (5' UR) mutant, *Cidec* and Synthetic PPRE reporters using QuickChange mutagenesis kit (Stratagene) following the instructions provided by the manufacturer. The *Cidec* 5'UR mutant reporter was generated in the same way, based on the *Cidec* reporter. Primer sequences are given in Supplementary Table 1. The reporter construct 5xGal4-E1BTATA-pGL3 has previously been described[39]. The *Cidec* (promoter)-pGL3 luciferase reporter[40] was a kind gift of Dr. P.F. Marrero. pCDNA3.1 expression vectors for hPPARγ2 and hRXRα[17] were used to generate the hPPARγ2 mutants E379K, R212Q and R212W and the hRXRα K175E mutant, respectively, using the QuickChange mutagenesis kit (Stratagene). Gal4DBD-hPPARγ-AF2 WT, pGEX-PPARγ-LBD WT, pGEX-RXRα, pGEX-SMRT, and pGEX-SRC1 have been described previously[17,41] and mutants thereof were generated as described above.

For luciferase reporter assays U2OS and HEK293T cells were seeded in 24-wells plates and transiently transfected using PEI. Each well was cotransfected with a reporter construct (1 µg), 2 ng Renilla, and PPAR and/or expression constructs (10 ng). The next day, media were removed and fresh media were added with or without 1 µM Rosiglitazone, as indicated in the figure legends. After incubation, cells were washed twice with phosphate-buffered saline (PBS) and harvested in lysisbuffer (Promega), and assayed for luciferase activity according to the manufacturer's protocol (Promega Dual-Luciferase Reporter Assay System) and for Renilla to correct for transfection efficiency. The relative light units were measured by a Centro LB 960 luminometer (Berthold Technologies, Bad Wildbad, Germany). The results are averages of at least three independent experiments assayed in duplicate ±SEM. To compare three or more groups, an ordinary one-way analysis of variance (ANOVA) was performed with a Tukey's multiple comparison test to compare the mean of each group with that of every other group. A statistically significant difference was defined as a $p$-value <0.05.

## Western blot analysis

Protein expression of the transfected U2OS and HEK293T cells and of adenovirally transduced PPARγ$^{-/-}$ MEF-CARs was determined by western blotting. For this, lysates were boiled in Laemmli sample buffer for 5 min at 95 °C. Samples were subjected to SDS-PAGE and transferred to a Millipore membrane (Millipore). Anti-PPARγ (sc-7196; RRID: AB_654710), anti-Gal4 DBD (sc-510; RRID: AB_627655), anti-FABP4 (sc-18661; RRID: AB_2231568), anti-RXRα (sc-553; RRID: AB_2184874), anti-tubulin (Sigma−Aldrich T9026; RRID: AB_477593), anti-LPL[42], anti-FLAG-HRP (Sigma−Aldrich A8592; RRID: AB 439702), anti-HA (ab9110; RRID:AB_307019), anti-LgBiT (N710A) were used for detection of the proteins, all at dilution 1:1000 except for the anti-Gal4 DBD antibody which was diluted 1:500. Enhanced chemiluminescence (Amersham Biosciences) was used for visualization. Quantification of band intensity was performed with reference to loading control (tubulin) and PPARγ using the ImageJ Gel Analysis program.

## DNA affinity purifications

Lentiviral vectors for stable expression of PPARγ in U2OS cells were generated by replacing beta-catenin with the PPARγ2 expression cassette in pLV-CMV-FLAG-betaCatenin-Ires-PURO, a kind gift of Dr. J. de Rooij. Lentiviral particles were produced in HEK293T cells. After lentiviral transduction, stably transduced U2OS cells were selected and maintained on 2 μg/ml puromycin. Nuclear extracts from U2OS were prepared as described previously[43]. Oligonucleotides containing either the *CIDEC* wt or *CIDEC* dead motif (Integrated DNA Technologies) with 5′ biotinylation of the forward strand (Supplementary Table 1) were annealed using a 1.5× molar excess of the reverse strand. For DNA affinity purifications[43], 500 pmol of DNA oligonucleotides were immobilized using 20 μl of Streptavidin-Sepharose bead slurry (GE Healthcare, Chicago, IL). Then, 500 μg of nuclear extract and 10 μg of non-specific competitor DNA (5 μg polydAdT, 5 μg polydIdC) were added to each pulldown. After extensive washing, samples were prepared for mass spectrometry analysis or western blotting.

## Mass spectrometry analysis

For mass spectrometry analysis, beads were resuspended in elution buffer (2 M urea, 100 mM TRIS (pH 8.0), 10 mM DTT) and alkylated with 50 mM iodoacetamide. Proteins were digested on beads with 0.25 μg of trypsin for 2 h. After the elution of peptides from beads, an additional 0.1 μg of trypsin was added, and digestion was continued overnight. Peptides were labeled on Stage tips using dimethyl labeling[43]. Each pulldown was performed in duplicate and label swapping was performed between duplicates to avoid labeling bias. Matching light and heavy peptides were combined and analyzed on an Orbitrap Exploris (Thermo) mass spectrometer with acquisition settings described previously[44]. RAW mass spectrometry data were analyzed with MaxQuant 1.6.0.1 by searching against the UniProt curated human proteome (released June 2017) with standard settings. Protein ratios obtained from MaxQuant were used for outlier calling. An outlier cutoff of 1.5 interquartile ranges in two out of two replicates was used. Western blot analysis of eluted proteins was performed using the FLAG epitope tag for detection.

## Modeling of PPARγ-RXRα 3D structures

To analyze the impact of the E379K mutation in PPARγ, which is located in a functionally less well-characterized region than the R212Q mutation in the hinge region, we used the HADDOCK2.2 web server[45]. As the starting point for the modeling we used the crystal structure of the intact PPARγ:RXRα nuclear receptor complex with DNA (PDB ID 3DZY) (isoform 1)[11]. Both WT and E379K (E351K isoform 1) mutant structures were subjected to a short refinement in explicit solvent using the refinement interface of the HADDOCK server. The mutation was introduced by changing the residue name in the PDB file and letting HADDOCK rebuild any missing atoms.

## Peptide analyses by NMR and CD spectroscopy

Peptides were purchased from TAG Copenhagen A/S (Denmark), purified to ≥ 95% by reversed-phase HPLC and identities confirmed by mass spectrometry.

For circular dichroism (CD) spectroscopy, samples of 100 μM WT-PPARγ$^{376-385}$ and E379K- PPARγ$^{376-385}$ peptides were prepared in 20 mM Na$_2$HPO$_4$ /NaH$_2$PO$_4$ (pH 7.4). Far-UV CD spectra of the peptides and of a matched buffer sample were recorded from 260 to 190 nm on a Jasco J-810 spectropolarimeter in a 0.1 cm Quartz cuvette (Hellma, Suprasil®) with a Peltier controlled temperature set to 25 °C. Data acquisition was carried out with a scanning speed, 10 nm/min; data pitch, 0.1 nm; accumulations, 10; bandwidth, 1 nm; response time, 2 s. After subtraction of the buffer spectrum, the spectra were smoothed by Fast Fourier Transformation. Helical percentage was calculated using (%H = ([θ]$_{222}$ − 3000)/(−36,000 − 3000)) essentially as described[46].

To assign the resonances of the WT-PPARγ$^{376-385}$ and E379K-PPARγ$^{376-385}$ peptides, NMR experiments were recorded on 2.3 mM PPARγ$^{376-385}$ in 20 mM Na$_2$HPO$_4$ /NaH$_2$PO$_4$ (pH 7.4) at 25 °C on a Bruker Avance NEO 800 MHz spectrometer equipped with a 5 mm CPTXO Cryoprobe C/N·H·D. The following spectra were acquired at natural isotope abundance: 1D $^1$H (zgesgp), $^1$H -$^{15}$N HSQC (hsqcetfpf3gp), $^1$H–$^{13}$C HSQC (hsqcetgpsisp2.2), 2D TOCSY (dipsi2esgpph, mixing time 80 ms) and 2D ROESY (roesyesgpph, mixing time 250 ms). The spectra were transformed using TopSpin 3.6.2 and analyzed manually in CcpNmr Analysis 2.5.0[47]. All chemical shifts were referenced to internal DSS directly ($^1$H) or indirectly ($^{13}$C, $^{15}$N) as described[48]. Secondary chemical shifts of Cα were calculated using the random coil set from Kjaergaard et al.[49].

## NR-Coregulator Interaction analyses

Rosetta pLysS competent bacteria (Novagen, EMD Chemicals Inc., Darmstadt, Germany) were transformed with GST expression plasmids. GST-fusion protein expression was induced with 1 mM IPTG and purified on Glutathione-Sepharose beads (Amersham Biosciences)[39].

For NR-Coregulator interaction profiling, GST-fusion proteins were eluted from gluthatione-sepharose beads (Amersham, Buckinghamshire, UK) using elution buffer (20 mM glutathione, 100 mM Tris pH 8.0, 120 mM NaCl). Proteins were concentrated using Vivaspin centrifugal concentrators (Sartorius, Epsom, UK) and protein concentrations were determined using SDS-PAGE followed by Coomassie Brilliant Blue staining. Assay mixes were prepared on ice in a master 96-Well plate, with 5 nM of purified PPAR-γ LBD-GST WT and E379K mutant (see below), TR-FRET Coregulator buffer F (Invitrogen), 25 nM Alexa488-conjugated anti-GST antibody (A11131, Invitrogen), 5 mM DTT, 2% DMSO with or without 1 mM Rosiglitazaone. All assays were performed in a PamStation®−96 controlled by EvolveHT software (PamGene International BV, 's-Hertogenbosch, The Netherlands) at 20 °C, at a rate of 2 cycles per minute. Nuclear Receptor PamChip® Arrays (PamGene International BV, 's-Hertogenbosch, The Netherlands) contained 53 peptides[21]. Per array, 25 μl of assay mix was transferred from the master plate to the chip using a multichannel pipette. During the period of ligand incubation (~40 minutes), a solution of GST-PPARγ-LBD, fluorescent anti-GST antibody, and ligand was pumped through the porous peptide-containing membrane for 81 cycles at a rate of 2 cycles per minute. Assay mixes were incubated in the arrays for 80 cycles and a tiff format image of every array was obtained at cycles 21, 41, 61, 81 by a CCD camera-based optical system integrated in the PamStation®−96 instrument.

For GST pull-downs, GST or GST-fusion proteins were incubated with 10 μl of rabbit reticulolysate containing translated $^{35}$S−labeled protein (TNT T7 Quick Coupled Transcription/Translation Kit, Promega) for 3 hours at 4 °C. After incubation, the beads were washed three times with NETN-buffer and Laemli sample buffer was added.

Samples were boiled at 95 °C for 5 min and loaded on a 10% SDS-polyacrylamide gel. Proteins were stained with Coomassie Brilliant Blue, and gels were fixed (20% methanol and 10% acetic acid for 10 min) and dried. Radioactive signals were analyzed with a Storm 820 Imager (Molecular Dynamics).

## Protein complementation assays (PCA)

Heterodimerization between PPARγ2 and hRXRα was analyzed in live cells with the NanoBiT® PPI System (Promega)[50]. To generate a protein fusion pair displaying effective complementation of the split luciferase, the Large BiT (LgBiT; 18 kD) of luciferase was fused to the C-terminus of PPARγ2, while the Small BiT (SmBiT; 1.3 kDa) was fused to the N-terminus of hRXRα (Supplemental Fig. 3a). Mutants were generated using the QuickChange mutagenesis kit (Stratagene). Mutants previously shown to have partially or completely lost heterodimerization capacity (R425C and L464R, respectively[17]) were included as controls.

HEK293T cells were seeded in 96-well plates and transfected with plasmids encoding PPARγ2-LgBiT (50 ng/well) or mutant versions, and smBiT-hRXRα (50 ng/well) using Xtreme gene (Sigma–Aldrich). After 48 h, cells were washed with PBS and diluted substrate (Nano-Glo® Live Cell Assay System; Promega) was added according to the manufacturer's protocol. The relative light units were measured by a Centro LB 960 luminometer (Berthold Technologies, Bad Wildbad, Germany). The results are averages of at least three independent experiments assayed in triplicate ±SEM. To compare three or more groups, an ordinary one-way analysis of variance (ANOVA) was performed with a Tukey's multiple comparison test to compare the mean of each group with that of every other group. A statistically significant difference was defined as a $p$-value <0.05. After luminescence analysis, cells were lysed in RIPA buffer, triplicates were pooled and subjected to Western blot analysis, as described above.

## Adenovirus generation and purification

Recombinant adenoviruses containing a hemagglutinin (HA)-tagged mouse PPARγ2 were generated using AdEasy cloning system (Stratagene)[3]. The Ad-HA-mPPARγ2-WT[6] was used as a template to generate the mutations R212Q and E379K using QuickChange mutagenesis kit (Stratagene) following the instructions provided by the manufacturer. Successful mutagenesis was verified by Sanger sequencing. The plasmids were linearized and transfected into the HEK293 cell line for amplification and purification using CsCl gradients.

## Adenoviral delivery of PPARγ in PPARγ$^{-/-}$ MEF-CAR cells

PPARγ$^{-/-}$ MEF-CAR cells[6] were transduced with adenoviral vectors expressing HA-tagged mPPARγ2-WT, mPPARγ2-R212Q or mPPARγ2-E379K. After two hours of virus exposure, the medium was removed, and cells were treated with medium containing either vehicle dimethyl sulfoxide (DMSO) or 1 μM rosiglitazone (Alexis) for another 6 hours before harvest for RNA and protein expression analyses and for ChIP- and ATAC-seq.

## Differentiation assay

One-day post confluent PPARγ$^{-/-}$ MEF-CAR cells were transduced with adenoviral vectors expressing haemagglutinin (HA)-tagged mPPARγ2-WT or mutant PPARγ2. Following 2 h of virus exposure, the medium was removed and replaced by differentiation medium A (DMEM supplemented with 10% FBS (Gibco), 5 μg/mL insulin, 1 μM dexamethasone, 0.5 mM isobutylmethylxanthine, and 1 μM rosiglitazone (Alexis)). After three days, the cells were maintained in differentiation medium B (DMEM supplemented with 10% FBS (Gibco) and 5 μg/mL insulin). At day 7, differentiation was evaluated using the Oil-Red-O staining method[17].

## RNA analysis

RNA extraction, cDNA synthesis, and qPCR were performed as described previously[25]. For RNA-seq analysis, RNA from two independent experiments was prepared in triplicate. After qPCR validation, the triplicates were pooled and prepared for sequencing according to the manufacturer's instructions (Illumina).

Samples were sequenced on Illumina Novaseq 6000 using NovaSeq Control Software v1.3.0, and the quality of sequenced reads was assessed using FastQC (https://qubeshub.org/resources/fastqc). Reads were aligned to the mouse genome (mm10) using STAR[51] with default settings. HOMER makeTagDirectory[52] was used to generate tag directories for all conditions on primarily aligned reads. HOMER analyzeRepeats.pl was used to count reads within exons of the mm10 genome with the settings -count exons, -condenseGenes, -noCondensing, -noadj. DESeq2[53] was used to identify differentially regulated genes between conditions.

## ChIP-seq

ChIP was performed as previously described[6]. Samples were sonicated using the Covaris ME220 sonicator (Covaris, Woburn, MA, USA); peak power 75, duty factor 26.66%, 500 cycles/burst for 7.5 minutes. Antibodies directed against hemagglutinin (HA, Abcam Ab9110; RRID: AB_307019), H3K27ac (Abcam, Ab4729, RRID: AB_2118291), and MED1 (Bethyl Laboratories, A300–793A; RRID: AB_577241) were used for immunoprecipitation. For ChIP-seq, the ChIP reactions were scaled to obtain 10 ng DNA. After decrosslinking, the DNA that had undergone ChIP was prepared for sequencing according to the manufacturer's protocol (Illumina). ChIP-seq was performed on two independent experiments.

**Alignment and construction of tag directories.** ChIP-seq libraries were sequenced on the Illumina NovaSeq 6000 using NovaSeq Control Software v1.4.0 and v.1.6.0, and the quality of sequenced reads was assessed using FastQC (https://qubeshub.org/resources/fastqc). Reads were aligned to the mouse genome (mm10) using STAR[51] with settings: outFilterMismatchNmax 2, alignIntronMax 1, outSJfilterIntronMaxVsReadN 0, and outFilterMatchNmin 25. Picard Tools (Broad Institute http://broadinstitute.github.io/picard/) were used to deduplicate primary aligned reads. HOMER makeTagDirectory was used to generate tag directories for all conditions, allowing only one read per position per length of the read (-tbp 1). Due to the uneven sequencing depth of H3K27Ac ChIP-seq samples, one tag directory was down-sampled to 20 mill. reads.

**Identification of PPARγ WT binding sites.** Putative PPARγ binding sites were defined as follows using the HA-PPARγ ChIP-seq data. First, replicates from HA-PPARγWT treated cells were pooled. Homer findPeaks[52] was used to call putative PPARγ sites using settings -style factor, -localSize 20000, and the pooled tag directory from control samples as reference. Peaks were extended to 500 bp centered around the peak center. Finally, artifact regions were discarded based on the ENCODE Consortium blacklist for the mm10 genome[54]. Peaks with a tag count of fewer than 35 tags were filtered away.

**Identification of sites differentially bound by PPARγ, Med1, or acetylated at H3K27.** PPARγ, Med1 and H3K27Ac ChIP-seq tag counts were quantified in the above defined PPARγWT binding regions using HOMER annotatePeaks[52] with no normalization of sequencing depth (-noadj). Prior to counting H3K27Ac ChIP-seq tags the PPARγ binding sites were extended ±1.5 kb of the peak center to capture more histone ChIP-seq signal, as histone modifications are generally distributed broadly around transcription factor binding sites. Next, we used DESeq2[53] to (1) call sites that gain Med1 and/or H3K27ac signal upon PPARγWT expression (denoted PPARγ-target enhancers); (2) call sites that were differentially bound by PPARγ, Med1, or acetylated when

comparing WT and mutant PPARγ; and (3) obtain PPARγ, Med1, and H3K27Ac counts normalized to the total tag directory size.

**Motif analysis.** HA-PPARγWT binding sites were scanned for the PPRE motif from the JASPAR database[55] in a region of 200 bp around the peak center and the motif score was extracted using HOMER annotatePeaks[52]. If more than one PPRE motif was identified within a PPARγ-binding region, the motif with the highest motif score was used for further analyses.

De novo motif search was made using Homer findMotifsGenome[52] with motif length of 15–17 bases and searched within ±100 bp of peak center of PPARγ-target enhancers. The de novo motif was hereafter used to scan all PPARγ-binding sites using HOMER annotatePeaks.

To analyze the motif score within sections of the motif, position files of PPARγ-binding sites were used to extract the sequence from the FASTA sequence of the mm10 reference genome. The de novo motif was extracted within these sequences using Homer2 find[56], where the threshold for calling a motif was set to −2. The motif score was hereafter calculated within sections; 5′ extension (base 1–4), PPARγ-half site (base 5–10), and RXR-half site (base 12–17).

### ATAC-seq
ATAC-seq was made in two biological replicates. 100,000 cells were centrifuged $500 \times g$ for 5 min, 4 °C. The cells were washed once in 200 μl DPBS and resuspended in 50 μl tagmentation mix (33 mM Tris-acetate, pH 7.8, 66 mM potassium acetate, 10 mM magnesium acetate, 16% dimethylformamide, 0.01% digitonin and 2.5 μl Tn5 (Nextera, Illumina)[57]). Tagmentation was performed for 30 min at 37 °C, shaking at 800 rpm. Tagmented DNA was purified using Qiagen PCR purification kit according to the protocol of the manufacturer. Purified tagmented DNA was prepared for sequencing using dual-unique index primers (Illumina TG Nextera® XT Index Kit) and Q5 High fidelity DNA polymerase (NEB) under the following PCR conditions: 72 °C 5 min; 98 °C 1 min; 10 cycles 98 °C 10 s, 63 °C 30 s, 72 °C 20 s; 10 °C hold, purified in accordance with the protocol of the manufacturer (Illumina) and sequenced on NovaSeq 6000 using NovaSeq Control Software v.1.6.0.

Quality of sequenced reads was assessed using FastQC (https://qubeshub.org/resources/fastqc). Paired-end 50 bp reads were aligned to the mouse genome (mm10) using STAR[51] with settings: outFilterMismatchNmax 2, alignIntronMax 1, outSJfilterIntronMaxVsReadN 0, and outFilterMatchNmin 25. Primarily aligned reads were deduplicated using Picard Tools (Broad Institute http://broadinstitute.github.io/picard/) whereafter fragments of length ≤ 120 bp were selected for further analyses (nucleosome-free regions) using SAMtools[58]. Tag directories were generated using HOMER makeTagdirectories[52] Using HOMER analyseRepeats[52] ATAC-seq signal was counted within PPARγ-WT peaks extended ±250 bp around peak center and in bins of 50 bp in a window of 3000 bp centered at the peak center.

### Enrichment analysis
The number of enhancers in the vicinity of PPARγ-target genes relative to the number of enhancers in the vicinity of non-PPARγ-target genes was determined using BedTools[59]. Two hundred genes not regulated by PPARγ were randomly selected 10 times, and the average number of enhancers was used for normalization.

### Reporting summary
Further information on research design is available in the Nature Portfolio Reporting Summary linked to this article.

## Data availability
The sequence datasets generated in this study have been deposited at NCBI GEO under accession code GSE199426. The mass spectrometry proteomics data have been deposited to the ProteomeXchange Consortium via the PRIDE partner repository with the dataset identifier PXD036589. Processed data used in the preparation of this manuscript are detailed in the Source Data files provided with the manuscript. There are no restrictions on data access. Source data are provided with this paper.

## Code availability
Codes and scripts used to process and analyze data have been deposited to GitHub: https://github.com/mstahlmadsen/PPARgamma-lipodystrophy-mutants-reveal-intermolecular-interactions-required-for-enhancer-activation.

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

## Acknowledgements

This work was supported by grants from the Independent Research Fund Denmark (Sapere Aude Advanced grant no. 12-125524 to S.M.), the Danish National Research Foundation (DNRF grant no. 141 to the Center for Functional Genomics and Tissue Plasticity (ATLAS) (to S.M.)), the Novo Nordisk Foundation (support to S.M. through the NNF Center for Stem Cell Biology (NNF17CC0027852) and to BBK through the NNF Challenge center REPIN (NNF18OC0033926) and cOpenNMR (NNF18OC0032996), (the Villum Foundation (support to S.M. through the Villum Center for Bioanalytical Sciences and to K.N.N through grant VYI25397). The Vermeulen lab is part of the Oncode Institute, which is partly funded by the Dutch Cancer Society. We thank members of the Kalkhoven, Mandrup and Van Mil laboratories for helpful discussions and Dr. Wilko Spiering for clinical assessment. We thank Dr. Natasja Rochel and Dr. Bruno Klaholz for kindly providing the SAXS data of the DNA-bound PPARγ/RXRα heterodimer structure, Marijke Baltissen and Dr. Cornelia G. Spruijt for expert technical advice, Dr. Johan de Rooij for the pLV-CMV-FLAG-betaCatenin-Ires-PURO vector, and Dr. A. Bensadoun for kindly providing the antibody against mouse LPL.

## Author contributions

M.S.M., M.F.B., M.R.M., A.K., A.B., C.G., E.G.K.T., D.W., M.E.G.K., M.G.K., N.H., and J.M.R.P. designed and performed experiments and analyzed the data, M.S.M. and M.R.M. performed computational analyses, A.M.J.J.B. and B.K. performed detailed protein structure analyses, H.M. and D.C. performed patient identification and detailed clinical assessment, E.K., S.M., S.K., F.C.P.H., H.M., S.W.C.M., M.V., B.K., D.C., and K.N.N. supervised the study, M.S.M., M.F.B., S.M., and E.K. wrote the manuscript. S.M. and E.K. were responsible for the overall project management and co-supervised the research.

## Competing interests

The authors declare no competing interests.

## Additional information

¹Functional Genomics and Metabolism Research Unit, Department of Biochemistry and Molecular Biology, University of Southern Denmark, Odense, Denmark. ²Center for Molecular Medicine, University Medical Center Utrecht, Utrecht University, Utrecht, The Netherlands. ³Department of Molecular Biology, Faculty of Science, Radboud Institute for Molecular Life Sciences, Oncode Institute, Radboud University Nijmegen, 6525 Nijmegen, GA, The Netherlands. ⁴Department of Biology, University of Copenhagen, Copenhagen, Denmark. ⁵Faculty of Science—Chemistry, Bijvoet Centre for Biomolecular Research, Utrecht University, Utrecht, The Netherlands. ⁶Division of Human Nutrition, Nutrition, Metabolism and Genomics Group, Wageningen University, Wageningen, The Netherlands. ⁷Internal Medicine, Rijnstate Hospital, Arnhem, The Netherlands. ⁸Department of Vascular Medicine, University Medical Center Utrecht, Utrecht University, Utrecht, The Netherlands. ⁹Center for Metabolic Diseases, Leuven University Hospitals, Leuven, Belgium. ¹⁰Present address: Department of Human Genetics, Amsterdam UMC, Vrije Universiteit Amsterdam, Amsterdam, The Netherlands. ¹¹Present address: Gubra, Hørsholm, Denmark. ¹²Present address: Princess Máxima Center for Pediatric Oncology, Utrecht, the Netherlands. ¹³Present address: Institute of Biochemistry and Center for Molecular Biosciences Innsbruck, University of Innsbruck, Innsbruck, Austria. ¹⁴These authors contributed equally: Maria Stahl Madsen, Marjoleine F. Broekema. ¹⁵These authors jointly supervised this work: Susanne Mandrup, Eric Kalkhoven. ✉e-mail: s.mandrup@bmb.sdu.dk; e.kalkhoven@umcutrecht.nl

