## [Peer Review File · Nature Communications]

PPAR γ lipodystrophy mutants reveal intermolecular interactions required for enhancer activationREVIEWER COMMENTS

Reviewer #1 (Remarks to the Author):

The manuscript by Stahl et al. should be of significant interest to the PPARg and human lipodystrophy arena, and contains a wealth of carefully conducted and well-rounded experiments and presents important findings related to human disease. The studies show that missense mutations that occur precisely at sensitive functional surfaces of the PPARg protein lead to significant functional changes and more specifically to the ability of PPARg to bind to many enhancer sites within chromatin. One of these mutations (R212Q) interferes with the ability to bind to a critical DNA-sequence often seen just upstream of the direct-repeat -1 (DR-1) elements characteristic of PPARg binding sites. The 2nd mutation (E379K) lies in an interface that appears to critically guide the interaction of the PPARg LBD with the RXR DBD when their complex is bound to the extended DR-1. The writing is for the most part clear and the methodology is sound and meets the expected standards of this journal. The work is highly original and the results are noteworthy. Therefore, I am highly supportive for the publication of the manuscript, but with the revisions below considered.

However, there are a few parts of the manuscript discussing previous structural studies that trouble me as they are over-simplistic and could more accurately and comprehensively present all the structural work on the PPARg-RXR/DR1 to date.

1) In several places within the manuscript, the authors seem to be saying that there is an X-ray structure (Chandra) and a SAX structure (Osz/Rochel). To be sure there is a single X-ray structure, but two different and contradictory SAX studies, and a great deal of other work too that is critical to the sound understanding of the importance of the closed structure. The authors should clarify which SAX structure they mean. There is the Osz/Rochel structure of PPARg/RXR/DR1 that reports an open structure. But there is another SAX structure published (A. Bernardes et al. PLOS 1, 2012) of the same PPARg/RXR/DNA complex that reports a closed structure in the DR1-bound form, fully in-line with the X-ray (Chandra) form and inconsistent with the SAX interpretation by Osz/Rochel. Also, the Chandra X-ray structural findings, especially the DBD-LBD contacts observed, were supported in the same publication with hydrogen-deuterium (H/D) exchange mass-spectrometry data and mutagenesis work. By contrast, there has been no independent verification of the open form, reported by Osz/Rochel to my knowledge. Moreover the other Osz/Rochel full-length structure discussed in their paper was that of RXR-RAR/DR1 and those findings were subsequently shown irreconcilable with the crystal structure, H/D ex studies, and mutagenesis studies on the same RXR/RAR/DR1 complex by others (see PMID: 29021580 Figure 5). These elements of the original Rochel paper's conclusions are worth pointing out for completeness, since the SAXS studies used in both PPAR/RXR and RAR/RXR cases were likely flawed or misinterpreted the same way in that paper.

2) In the Abstract 2nd sentence, the “structural studies indicate...” should be changed to state more specifically which types of structural studies. This could be remedied by specifying that “X-ray crystallographic” since the SAX studies by Osz and Rochel did not reveal any DBD-LBD or DBD-hinge interactions whatsoever.

3) Further to the above point, the Supp Figure 1 should be brought into the main Figure 1, and used to clearly contrast the Rochel/Osz SAXS structure with the X-ray structure while pointing to the location of the lipodystrophy mutations in each case. The legend should also make clear that there is the Bernardes closed SAX structure in-line with the crystal structure too.

4) Page 4, the finding that PPAR γ binds to the 5' half-site and its extension, are correctly linked to ref 3, but could the authors please check if references 4 and 5 are appropriately placed, since they are decades later than the original Ijpenberg report on polarity.

5) Page 4, line 86-90. To say that both X-ray and SAX showed several intermolecular interfaces and then cite in bulk the references 10-12 for that sentence is quite troubling. The authors should be more careful about the structural distinctions here, and even which SAX study they are discussing. As it is currently written, one may think that the X-ray and Rochel SAX studies reveal the same critical findings about intermolecular interfaces, which is incorrect.

6) On page 4 line 95, again authors need to clarify there are two SAXS studies with completely different conclusions. One is open and the other is closed, even though both studied the same PPAR γ /RXR/DNA ternary complex. Both cannot be right at the same time of course, so this is worth pointing out. So this so-called controversy is also not about solution state vs. crystal state or open vs. closed, as this kind of oversimplification has muddied the waters for many years.

7) Also page 4 line 95: “In contrast, SAXS.... heterodimerization mainly involves...”. Should be changed to “heterodimerization ONLY involves the two LBDs” since that is the only contact region between RXR and PPAR that the Osz group’s structure included. Even so, the heterodimerization between LBDs of nuclear receptor was shown by others well before the Rochel paper, and this LBD-LBD did not so much emerge as a new finding from their SAX studies, but as in-going assumption of the model they attempted to fit into their SAX data plots.

8) At the end of the discussion, where the authors underscore the importance of intermolecular interactions in PPAR functions such as between the LBD to DBD, it is worth noting that LBD-DBD interactions have now become apparent in all four of the nuclear receptor full-length/multi-domain structures reported to date, and not just in the RXR-PPAR structure (see PMID: 29021580, Figure 4). Therefore, there is good reason in the future to look for disease-causing mutations at these sensitive and critical domain-domain interfaces.

Reviewer #2 (Remarks to the Author):

Madsen et al. identified and studied two previously uncharacterized lipodystrophy mutations at the level of the nuclear receptor PPAR γ , known to be involved in adipogenesis. These mutations were predicted by the authors to interfere with two different interaction interfaces of PPAR γ , and the study went on to characterize the gene regulatory capacity of the mutants at a genome wide level and elucidated the functional consequences of the mutant compared to wt receptor activity making use of an elegant model system of stably transduced immortalized PPAR γ -/- mouse embryonic fibroblasts. The data are solid, well explained and are nicely underbuilding the hypotheses. The study is insightful and expected to be highly impactful to the field given that the way the PPAR γ LBD-RXR α DBD heterodimerization interface is most likely formed, has been a topic of debate, since many years. Intuitively, it seemed difficult at first to comprehend that a number of more than 4-fold enhancers are selectively less bound by E379K as compared to R212Q (with the latter mutant seemingly in the better position to interfere with chromatin), but when the study progressed, the authors solved the mechanism and found a plausible explanation, by revealing that esp. the E379K mutation has the greatest impact on binding of PPAR γ to both inaccessible and accessible target sites in chromatin, in line with a more general destabilization of the PPAR γ :RXR heterodimer by this mutant.

I do not have major comments, only some minor comments to consider should the authors find it be possible to address and clarifications that may be helpful. The one slight weak point, given there does seem an involvement of clinicians in this study, is that the study does not contain a closing piece that illustrates that some of the target genes are indeed affected more than others in relevant patient material. However, this is just a minor point and does not compromise the validity of the findings in this study, which still makes significant contributions towards solid molecular mechanisms as an essential leap forward to understand how different PPAR γ mutants can cause lipodystrophy.

1) Authors present an elegant model to explain the data. Both mutations, even though considerably apart in distance and positioned in different receptor domains, have a strong impact on differentiation capacities and place a significant burden on PPAR γ -induced gene activation, which is chosen as a defensible focus point in the study considering the increased likelihood of a direct mechanism of gene regulation. Understandably, authors focus in Supplementary Figure 2 only on the E379K mutant to explore the inherent transcriptional capacity in a Gal4 experiment and to investigate coregulator binding in a GST pull-down and coregulator array assay. I wonder, given the many recent studies on nuclear receptor allostery and the impact of mutations in domains different from the LBD on the function of those LBD domains, whether it is then still logic to assume the R212 mutant also will have no impact at

all on coregulator associations? Have authors checked this by any chance? Could it for instance be, as an addition to the mechanisms presented, that the R212 mutant has a tendency to favor corepressors over coactivators?

2) In Supp Figure 2D the necessary normalization controls that verify that the arrays have been captured for similar amounts of time should also be shown and indicated with a circle.

3) Can authors present also a normalized quantification of Figure 3D? Because the mutant levels are substantially lower here when compared to wt PPAR γ , a normalization and presentation of ratio's (e.g. via Image J) would be helpful here.

4) Line 186 mentions a "data not shown", perhaps this can be included still in the Supp Info?

5) Line 140 typo, should read: "interface"

6) Line 151 typo, should read: "displays"

7) Line 230 states that "the R212Q mutant more dramatically affects the ability of PPAR γ to acutely activate target genes". I am unsure what is meant with "acutely". Does this comprise also the time-dependent component of the differentiation process? Please clarify.

8) Supp Figure 6 typo, should read: "dependent"

Reviewer #3 (Remarks to the Author):

In this manuscript by Madsen et al., the authors identified two previously uncharacterized PPAR gamma (PPAR γ) mutations that result in familial lipodystrophy. The authors showed that, molecularly, these mutations affect the interaction between PPAR γ and its cognitive binding motif at the 5' extension region (R212Q) or between PPAR γ and RXR heterodimers (E379K), resulting in slightly differential reduction in the activation of PPAR γ target genes due to the mutants' inability to engage with weak PPREs or inaccessible chromatin. Based on these results, the authors made two major claims: (a) they revealed that intermolecular interactions between PPAR γ /RXR/DNA are important for target gene activation in a cellular context; (b) subtle differences between mutants in target gene activation are determined by chromatin accessibility and PPRE motif strength.

However, the fact that RXR is an obligatory partner for PPAR γ activity and that subtle differences in the 5' extension of the PPRE motif determine PPAR γ -induced target gene expression are well-known (Mangelsdorf and Evans, Cell, 1995). It is thus not surprising that mutants that compromise these interactions reduce PPAR γ activities. Therefore the innovation in the current study is largely incremental. In addition, mutations at the R212 site have been shown to be pathological in causing lipodystrophy (Majithia et al. Nat Genet 2016).

Furthermore, while the molecular characterization of mutants is interesting, the experiments suffer from three main drawbacks: (1) they are conducted in a highly artificial cellular system (CAR-MEF) with overexpression of PPAR γ mutants. It is not clear the conclusions are all that relevant in physiology. A CRISPR-based HDR/prime editing to knock-in mutations in relevant cell types (e.g. mouse/human ADSC) would be much more desirable to evaluate the functions of these mutants. (2) Based on the gene expression profile (Fig. 4G and H), it appears that the clinical outcome of lipodystrophy may be driven by classic PPAR γ target genes that are dual sensitive (e.g. FABP4, CIDEC). It is not clear whether these two mutations differ in any way functionally (with regard to adipose tissue) and clinically (with regard to phenotypical presentation and severity), and compared to previously characterized mutants. This lack of specificity may stem from the use of the artificial MEF system, highlighting the need to characterize them in adipocytes. As such, and consistent with point (1), the relevance of molecular details between mutants is unclear. (3) Molecularly, while the authors showed the reduced ability of R212Q mutation to remodel chromatin, it is not clear how this is achieved. Which chromatin remodeling complexes are relevant to this observation?

Taken together, the main claim of the current study is largely incremental based upon two-decade old knowledge. It offers few details regarding the functional specificity (and its determinant) beyond what is known of how PPAR γ works, despite major technological advancements in the field which are widely available.

Point-by-point response to reviewers' comments

"PPAR γ lipodystrophy mutants reveal intermolecular interactions required for enhancer activation", Stahl Madsen, Broekema et al. nNCOMMS-22-09854-T

Reviewer #1 (Remarks to the Author):

The manuscript by Stahl et al. should be of significant interest to the PPAR γ and human lipodystrophy arena, and contains a wealth of carefully conducted and well-rounded experiments and presents important findings related to human disease. The studies show that missense mutations that occur precisely at sensitive functional surfaces of the PPAR γ protein lead to significant functional changes and more specifically to the ability of PPAR γ to bind to many enhancer sites within chromatin. One of these mutations (R212Q) interferes with the ability to bind to a critical DNA-sequence often seen just upstream of the direct-repeat -1 (DR-1) elements characteristic of PPAR γ binding sites. The 2nd mutation (E379K) lies in an interface that appears to critically guide the interaction of the PPAR γ LBD with the RXR DBD when their complex is bound to the extended DR-1. The writing is for the most part clear and the methodology is sound and meets the expected standards of this journal. The work is highly original and the results are noteworthy. Therefore, I am highly supportive for the publication of the manuscript, but with the revisions below considered.

We thank the reviewer for the positive assessment of our manuscript.

However, there are a few parts of the manuscript discussing previous structural studies that trouble me as they are over-simplistic and could more accurately and comprehensively present all the structural work on the PPAR γ -RXR/DR1 to date.

1) In several places within the manuscript, the authors seem to be saying that there is an X-ray structure (Chandra) and a SAX structure (Osz/Rochel). To be sure there is a single X-ray structure, but two different and contradictory SAX studies, and a great deal of other work too that is critical to the sound understanding of the importance of the closed structure. The authors should clarify which SAX structure they mean. There is the Osz/Rochel structure of PPAR γ /RXR/DR1 that reports an open structure. But there is another SAX structure published (A. Bernardes et al. PLOS 1, 2012) of the same PPAR γ /RXR/DNA complex that reports a closed structure in the DR1-bound form, fully in-line with the X-ray (Chandra) form and inconsistent with the SAX interpretation by Osz/Rochel. Also, the Chandra X-ray structural findings, especially the DBD-LBD contacts observed, were supported in the same publication with hydrogen-deuterium (H/D) exchange mass-spectrometry data and mutagenesis work. By contrast, there has been no independent verification of the open form, reported by Osz/Rochel to my knowledge. Moreover the other Osz/Rochel full-length structure discussed in their paper was that of RXR-RAR/DR1 and those findings were subsequently shown irreconcilable with the crystal structure, H/D ex studies, and mutagenesis studies on the same RXR/RAR/DR1 complex by others (see PMID: 29021580 Figure 5). These elements of the original Rochel paper's conclusions are worth pointing out for completeness, since the SAXS studies used in both PPAR/RXR and RAR/RXR cases were likely flawed or misinterpreted the same way in that

paper.

We thank the reviewer for these insightful comments and have now expanded the text comparing the X-ray structure (Chandra) and the SAXS structure (Osz/Rochel). Specifically, we have included the paper by Bernades et al. (PMID 22363753) (l. 194-112 and l. 137-140) as suggested by the reviewer. We are aware of the controversy between the different approaches and their interpretations and have tried to explain this as objectively as possible. Importantly, our work is consistent with but does not prove the DBD-LBD contacts.

Prompted by this comment and the results in the paper by Bernardes et al, indicating that the PPAR γ : RXR α LBD:DBD interface can only be observed in the DNA-bound state, we analyzed the effect of the E379K mutation on heterodimerization in the absence of DNA binding. Our analyses support this view, as we see that the E379K mutation does not interfere with heterodimerization in the absence of DNA. These data are now included as Supplementary Fig. 2 and mentioned in the Results section (l. 188-191). Furthermore, we have edited the model figure in Fig. 8D to illustrate this.

2) In the Abstract 2nd sentence, the “structural studies indicate...” should be changed to state more specifically which types of structural studies. This could be remedied by specifying that “X-ray crystallographic” since the SAX studies by Osz and Rochel did not reveal any DBD-LBD or DBD-hinge interactions whatsoever.

As the reviewer pointed out above, the SAXS studies by Bernardes et al., which we had not cited properly in the previous version, support the X-ray crystallographic studies by Chandra et al. As we now describe the Bernardes paper in the Introduction (l. 94-112), we consider it most appropriate to keep the wording “structural studies...” in the Abstract (l. 42).

3) Further to the above point, the Supp Figure 1 should be brought into the main Figure 1, and used to clearly contrast the Rochel/Osz SAXS structure with the X-ray structure while pointing to the location of the lipodystrophy mutations in each case. The legend should also make clear that there is the Bernardes closed SAX structure in-line with the crystal structure too.

In Supplementary Fig. 1 we show the Rochel/Osz SAXS structure and have now indicated the lipodystrophy mutants, as suggested by the reviewer. As our data, in particular those on the E379K mutant and the compensating mutation in RXR (Fig. 2), align better with the X-ray structure than the SAXS structure by Osz/Rochel, we feel that keeping the SAXS structure as a Supplementary figure is justified. As mentioned above, we have expanded the text comparing the X-ray structure (Chandra) and the SAXS structure (Osz/Rochel) and also specifically included the Bernardes paper (Introduction, (l. 194-112), as suggested by the reviewer. We have also mentioned the paper by Bernades et al. in the legend to Fig. 1D (l. 751).

4) Page 4, the finding that PPAR γ binds to the 5' half-site and its extension, are correctly linked to ref 3, but could the authors please check if references 4 and 5 are appropriately placed, since they are decades later than the original Ijpenberg report on polarity.

The reviewer is correct in that reference 3 (Ijpenberg et al.) dates back to 1997, while reference 4 and 5 are from 2008 and 2015, respectively. The paper by Ijpenberg specifically demonstrate the importance of the 5' extension of the core PPRE for a few selected natural PPREs. The more recent studies are genome-wide studies which have confirmed the 5' extension as an important for PPAR:RXR binding to DNA in chromatin. We have now split these references over 2 sentences, to indicate this more clearly (l. 69-70).

5) Page 4, line 86-90. To say that both X-ray and SAX showed several intermolecular interfaces and then cite in bulk the references 10-12 for that sentence is quite troubling. The authors should be more careful about the structural distinctions here, and even which SAX study they are discussing. As it is currently written, one may think that the X-ray and Rochel SAX studies reveal the same critical findings about intermolecular interfaces, which is incorrect.

We thank the reviewer for pointing this out and have adapted the text accordingly (l. 94-112).

6) On page 4 line 95, again authors need to clarify there are two SAXS studies with completely different conclusions. One is open and the other is closed, even though both studied the same PPARg/RXR/DNA ternary complex. Both cannot be right at the same time of course, so this is worth pointing out. So this so-called controversy is also not about solution state vs. crystal state or open vs. closed, as this kind of oversimplification has muddied the waters for many years.

We thank the reviewer for pointing this out and have adapted the text accordingly (l. 94-112).

7) Also page 4 line 95: "In contrast, SAXS.... heterodimerization mainly involves...". Should be changed to "heterodimerization ONLY involves the two LBDs" since that is the only contact region between RXR and PPAR that the Osz group's structure included. Even so, the heterodimerization between LBDs of nuclear receptor was shown by others well before the Rochel paper, and this LBD-LBD did not so much emerge as a new finding from their SAX studies, but as in-going assumption of the model they attempted to fit into their SAX data plots.

We agree with the reviewer that this point could have been phrased more explicitly and have adapted the manuscript accordingly (l. 94-112).

8) At the end of the discussion, where the authors underscore the importance of intermolecular interactions in PPAR functions such as between the LBD to DBD, it is worth noting that LBD-DBD interactions have now become apparent in all four of the nuclear receptor full-length/multi-domain structures reported to date, and not just in the RXR-PPAR structure (see PMID: 29021580, Figure 4). Therefore, there is good reason in the future to look for disease-causing mutations at these sensitive and critical domain-domain interfaces.

As the reviewer indicates, intermolecular interactions have indeed also been described in nuclear receptors other than PPAR-RXR. We have now mentioned this in the Discussion (l. 414-416), including the suggestion that disease-causing mutations may be found in these interfaces.

Reviewer #2 (Remarks to the Author):

Madsen et al. identified and studied two previously uncharacterized lipodystrophy mutations at the level of the nuclear receptor PPAR γ , known to be involved in adipogenesis. These mutations were predicted by the authors to interfere with two different interaction interfaces of PPAR γ , and the study went on to characterize the gene regulatory capacity of the mutants at a genome wide level and elucidated the functional consequences of the mutant compared to wt receptor activity making use of an elegant model system of stably transduced immortalized PPAR γ -/- mouse embryonic fibroblasts. The data are solid, well explained and are nicely underbuilding the hypotheses. The study is insightful and expected to be highly impactful to the field given that the way the PPAR γ LBD-RXR α DBD heterodimerization interface is most likely formed, has been a topic of debate, since many years. Intuitively, it seemed difficult at first to comprehend that a number of more than 4-fold enhancers are selectively less bound by E379K as compared to R212Q (with the latter mutant seemingly in the better position to interfere with chromatin), but when the study progressed, the authors solved the mechanism and found a plausible explanation, by revealing that esp. the E379K mutation has the greatest impact on binding of PPAR γ to both inaccessible and accessible target sites in chromatin, in line with a more general destabilization of the PPAR γ :RXR heterodimer by this mutant.

I do not have major comments, only some minor comments to consider should the authors find it be possible to address and clarifications that may be helpful. The one slight weak point, given there does seem an involvement of clinicians in this study, is that the study does not contain a closing piece that illustrates that some of the target genes are indeed affected more than others in relevant patient material. However, this is just a minor point and does not compromise the validity of the findings in this study, which still makes significant contributions towards solid molecular mechanisms as an essential leap forward to understand how different PPAR γ mutants can cause lipodystrophy.

We thank the reviewer for the positive evaluation of our work. The main goal of the current study was to unravel intrinsic properties of the PPAR γ protein, through the use of natural loss-of-function mutants. We do agree with the reviewer that testing target genes in patient material would have added an extra level to our manuscript. Unfortunately, we have been unable to receive permission for collecting adipose tissue from any of the patients involved. It should also be noted that subcutaneous adipose depots are dramatically reduced in FPLD3 patients, and sampling is very difficult.

1) Authors present an elegant model to explain the data. Both mutations, even though considerably apart in distance and positioned in different receptor domains, have a strong impact on differentiation capacities and place a significant burden on PPAR γ -induced gene activation, which is chosen as a defensible focus point in the study considering the increased likelihood of a direct mechanism of gene regulation. Understandably, authors focus in Supplementary Figure 2 only on the E379K mutant to explore the inherent transcriptional

capacity in a Gal4 experiment and to investigate coregulator binding in a GST pull-down and coregulator array assay. I wonder, given the many recent studies on nuclear receptor allostery and the impact of mutations in domains different from the LBD on the function of those LBD domains, whether it is then still logic to assume the R212 mutant also will have no impact at all on coregulator associations? Have authors checked this by any chance? Could it for instance be, as an addition to the mechanisms presented, that the R212 mutant has a tendency to favor corepressors over coactivators?

We agree with the reviewer that whilst the R212Q mutant is in a region known to interact with DNA, it may also affect ligand binding and/or cofactor binding. We have tested the R212Q mutant in the context of the Gal4DBD-LBD protein and added these data to the experiments previously represented in Supplementary Fig. 2B for the E379K mutant. In this assay, where transcriptional activity reflects ligand and/or cofactor binding, no significant differences were observed between the wt protein and the R212Q mutant. These data have now been included in an updated version of Supplementary Fig. 2B. The E379K mutant was tested in more detail (i.e. binding of selected cofactors and cofactor peptide array; Supplementary Fig. 2C and D), given its location in the LBD that is traditionally associated with ligand binding and cofactor binding. We would like to stress that these results cannot exclude effects of the R212Q mutation on ligand and co-factor binding in chromatin.

2) In Supp Figure 2D the necessary normalization controls that verify that the arrays have been captured for similar amounts of time should also be shown and indicated with a circle.

The original full images corresponding to Supplementary Fig. 2D (in revised form called Supplementary Fig. 3C), showing reference spots for image analysis, have now been included as Supplementary Fig. 12.

3) Can authors present also a normalized quantification of Figure 3D? Because the mutant levels are substantially lower here when compared to wt PPAR γ , a normalization and presentation of ratio's (e.g. via Image J) would be helpful here.

Normalized protein levels are now indicated in the legend to Fig. 3D (l. 787-789) and mentioned in the Materials and methods (l. 527-528).

4) Line 186 mentions a "data not shown", perhaps this can be included still in the Supp Info?

The data have now been included as Supplementary Fig. 1.

5) Line 140 typo, should read: "interface"

The typo has been corrected in the revised manuscript.

6) Line 151 typo, should read: "displays"

The typo has been corrected in the revised manuscript.

7) Line 230 states that “the R212Q mutant more dramatically affects the ability of PPAR γ to acutely activate target genes”. I am unsure what is meant with “acutely”. Does this comprise also the time-dependent component of the differentiation process? Please clarify.

We have included a reference to Fig. 4A, which represent the design of the experiment, to clarify that we refer to genes that are regulated within a 6-8h time frame (l. 259-260).

8) Supp Figure 6 typo, should read: “dependent”

The typo has been corrected in the revised manuscript.

Reviewer #3 (Remarks to the Author):

In this manuscript by Madsen et al., the authors identified two previously uncharacterized PPAR gamma (PPAR γ) mutations that result in familial lipodystrophy. The authors showed that, molecularly, these mutations affect the interaction between PPAR γ and its cognitive binding motif at the 5' extension region (R212Q) or between PPAR γ and RXR heterodimers (E379K), resulting in slightly differential reduction in the activation of PPAR γ target genes due to the mutants' inability to engage with weak PPREs or inaccessible chromatin. Based on these results, the authors made two major claims: (a) they revealed that intermolecular interactions between PPAR γ /RXR/DNA are important for target gene activation in a cellular context; (b) subtle differences between mutants in target gene activation are determined by chromatin accessibility and PPRE motif strength.

However, the fact that RXR is an obligatory partner for PPAR γ activity and that subtle differences in the 5' extension of the PPRE motif determine PPAR γ -induced target gene expression are well-known (Mangelsdorf and Evans, Cell, 1995). It is thus not surprising that mutants that compromise these interactions reduce PPAR γ activities. Therefore the innovation in the current study is largely incremental. In addition, mutations at the R212 site have been shown to be pathological in causing lipodystrophy (Majithia et al. Nat Genet 2016).

We agree with the reviewer that based on the physiological finding that the R212Q and E379K mutants give rise to lipodystrophy, one would expect these mutants to affect PPAR γ function. Furthermore, based on the X-ray structure, the R212Q and E379K mutations would be predicted to potentially interfere with the interaction of the hinge region with the DNA and the controversial interaction between the PPAR γ LBD and the RXR DBD, respectively. However, we respectfully disagree that our work to is incremental. The scope of our manuscript was not to show that these mutations affect PPAR γ function but rather to *how* they affect PPAR γ binding to DNA, co-factor recruitment and chromatin remodeling.

Our study is the first to investigate the molecular effects of PPAR γ lipodystrophy mutations at a genome-wide level. Our approach allows us to carefully characterize the molecular impact of the mutations in chromatin and to demonstrate novel roles of the predicted intermolecular interfaces. Specifically, we show for the first time:

- 1) The R212Q as well as the E379K reduce the adipogenic potential of PPAR γ but decrease transcriptional potential at only a subset of PPAR γ target genes.
- 2) Similarly, the mutations only affect the activation of a subset of PPAR γ -target enhancers. Interestingly, a major determinant of sensitivity to these mutations is low chromatin accessibility. Especially the R212Q mutation interferes with the ability of PPAR γ to recruit to and remodel target sites in nucleosome-rich chromatin, whereas there is little effect on binding to accessible nucleosome-free target sites. This indicates that the interaction between the PPAR γ hinge and the 5' extension of the PPRE is particularly important for the ability of PPAR γ to bind to target sites in nucleosome-rich chromatin.
- 3) The R212Q mutation furthermore interferes with chromatin remodeling capacity independent of PPAR γ binding, indicating that this interface may directly, or indirectly, e.g., via allosteric mechanisms, be involved in chromatin remodeling.
- 4) Our double charge reversal mutation assay, combining the E379K mutant with RXR α -K175E provide strong support for the existence and importance of the controversial interaction between the PPAR γ LBD and the RXR DBD. The finding from our genome-wide studies indicating that the E379K mutation also affects binding to accessible enhancers indicates that this interface is important for the general stabilization of the PPAR γ :RXR heterodimer on DNA.
- 5) Finally, our studies highlight how in-depth analyses of lipodystrophy mutants can unveil novel molecular mechanisms of PPAR γ function.

Furthermore, while the molecular characterization of mutants is interesting, the experiments suffer from three main drawbacks: (1) they are conducted in a highly artificial cellular system (CAR-MEF) with overexpression of PPAR γ mutants. It is not clear the conclusions are all that relevant in physiology. A CRISPR-based HDR/prime editing to knock-in mutations in relevant cell types (e.g. mouse/human ADSC) would be much more desirable to evaluate the functions of these mutants. (2) Based on the gene expression profile (Fig. 4G and H), it appears that the clinical outcome of lipodystrophy may be driven by classic PPAR γ target genes that are dual sensitive (e.g. FABP4, CIDEC). It is not clear whether these two mutations differ in any way functionally (with regard to adipose tissue) and clinically (with regard to phenotypical presentation and severity), and compared to previously characterized mutants. This lack of specificity may stem from the use of the artificial MEF system, highlighting the need to characterize them in adipocytes. As such, and consistent with point (1), the relevance of molecular details between mutants is unclear. (3) Molecularly, while the authors showed the reduced ability of R212Q mutation to remodel chromatin, it is not clear how this is achieved. Which chromatin remodeling complexes are relevant to this observation?

- 1) The MEF-CAR cellular system used in this study indeed relies on overexpression in a PPAR γ KO background, which has 3 important intrinsic advantages: i) it allows

unambiguous identification of PPAR γ -mediated effects (compared to empty vector control), ii) no interference of endogenous PPAR γ ; iii) it is a highly controlled system that allows identification of direct binding sites and direct target genes, avoiding secondary and tertiary effects of PPAR γ activation (both ChIP-seq and RNA-seq analyses can be performed within hours after transduction). Being able to determine the acute effects of PPAR γ wildtype and mutant expression without interference from endogenous PPAR γ is particularly important given the role of PPAR γ as a master regulator of adipocyte differentiation and function. CRISPR-based HDR/prime editing to knock-in mutations in mouse/human ADSC cellular systems would not enable us to distinguish direct from indirect effects of these mutations on target genes and target enhancers, in particular because of the differential effect on adipogenesis.

- 2) The clinical outcome of lipodystrophy may indeed be driven by classical PPAR γ target genes (e.g. FABP4, CIDEC) as the reviewer suggests, and CIDEC loss can be a cause of lipodystrophy (Rubio-Cabezas et al., 2009), but we have intentionally not drawn this conclusion in our manuscript, as we would see that as an overinterpretation of our data. Furthermore, the reviewer wonders whether the mutations we have characterized are different from previously described FPLD3-associated PPAR γ mutations. At a macroscopic level these mutations give rise to similar phenotypes as other FPLD3 mutations, but at a molecular level the mutations give rise to subtle and more selective defects than for example natural cysteine mutants that completely destroy the DBD (Agostini et al., 2006). Given the limited number of well-described FPLD3 cases it is currently too difficult to draw strong genotype-phenotype correlations. The current study was intended to specifically characterize two new mutations and shed light on the role of intermolecular interactions in the PPAR γ -RXR-DNA trimeric complex. Establishing such genotype-phenotype correlations will be a very exciting future project.
- 3) The reviewer is asking how the R212Q affects chromatin remodeling and which remodeling factor(s) are associated with the R212-dependent remodeling activity. This is an interesting point; however, we consider this beyond the scope of this work.

Taken together, the main claim of the current study is largely incremental based upon two-decade old knowledge. It offers few details regarding the functional specificity (and its determinant) beyond what is known of how PPAR γ works, despite major technological advancements in the field which are widely available.

As outlined above, we respectfully disagree with the reviewer. In our view this work provides novel mechanistic insight into the function of PPAR γ in chromatin. Our study is the first to investigate the molecular effects of PPAR γ lipodystrophy mutations at a genome-wide level. Our approach allows us to carefully characterize the molecular impact of the mutations in chromatin and to demonstrate novel roles of the predicted intermolecular interfaces.

REVIEWERS' COMMENTS

Reviewer #1 (Remarks to the Author):

I've read through the revised version, and see positive and constructive changes in response to my previous comments. The new version is of high quality and contains novel and well-supported findings of wide interest. I recommend its publication.

Point-by-point response to reviewers' comments

“PPAR γ lipodystrophy mutants reveal intermolecular interactions required for enhancer activation”, Stahl Madsen, Broekema et al. NCOMMS-22-09854A

Reviewer #1 (Remarks to the Author):

I've read through the revised version, and see positive and constructive changes in response to my previous comments. The new version is of high quality and contains novel and well-supported findings of wide interest. I recommend its publication.

We thank the reviewer for the positive assessment of our manuscript.